**EMBO** *reports*

# PAWS1 controls Wnt signalling through association with casein kinase 1α

Polyxeni Bozatzi[1,†], Kevin S Dingwell[2,†], Kevin ZL Wu[1], Fay Cooper[2], Timothy D Cummins[1], Luke D Hutchinson[1], Janis Vogt[1], Nicola T Wood[1], Thomas J Macartney[1], Joby Varghese[1], Robert Gourlay[1], David G Campbell[1], James C Smith[2,*] & Gopal P Sapkota[1,**]

## Abstract

The BMP and Wnt signalling pathways determine axis specification during embryonic development. Our previous work has shown that PAWS1 (also known as FAM83G) interacts with SMAD1 and modulates BMP signalling. Here, surprisingly, we show that overexpression of PAWS1 in *Xenopus* embryos activates Wnt signalling and causes complete axis duplication. Consistent with these observations in *Xenopus*, Wnt signalling is diminished in U2OS osteosarcoma cells lacking PAWS1, while BMP signalling is unaffected. We show that PAWS1 interacts and co-localises with the α isoform of casein kinase 1 (CK1), and that PAWS1 mutations incapable of binding CK1 fail both to activate Wnt signalling and to elicit axis duplication in *Xenopus* embryos.

**Keywords** BMP; CK1; FAM83G; PAWS1; Wnt
**Subject Category** Signal Transduction

## Introduction

PAWS1 (protein associated with Smad1) interacts with Smad1, an intracellular mediator of bone morphogenetic protein (BMP) signalling, and in particular, it modulates SMAD4-independent BMP activity [1]. The protein is conserved in vertebrates, and it belongs to the poorly characterised FAM83 family (FAMily with sequence similarity 83; FAM83A-H), which is defined by an N-terminal *DUF1669* domain of unknown function and contains a pseudo-phospholipase D catalytic motif [2]. Beyond the *DUF1669* domain, there is no sequence similarity between FAM83 members [1,3]. There are two known conditions mapped to mutations in FAM83G. In mice, the wooly mutation (*wly*), which results in a matted and rough coat in 3- to 4-week-old pups, is due to aberrant splicing of PAWS1 resulting in a severely truncated protein [4]. The second is a recessive missense mutation (p.R52P) within the DUF1669 domain of FAM83G in dogs suffering from hereditary footpad hyperkeratosis (HFH) [5]. The affected dogs develop hard and cracked footpads that over time become severely debilitating [5]. However, our knowledge of the molecular function of FAM83G/PAWS1 is limited.

BMP signalling plays important roles in development and in adult tissue homeostasis. In the developing *Xenopus* embryo, a gradient of BMP activity helps pattern the dorso-ventral axis, with the highest levels of BMP signalling promoting formation of the most ventral tissues [6,7]. In an effort to explore the function of PAWS1 in more detail, we overexpressed the protein in early *Xenopus* embryos. To our surprise, PAWS1 did not cause embryos to be ventralised but instead induced complete secondary axes, including well-formed heads. Such a response is typically obtained after ectopic activation of the Wnt signalling pathway [8], and we confirmed both in *Xenopus* and in U2OS osteosarcoma cells that PAWS1 does regulate Wnt signalling.

Mass spectrometric analysis revealed that PAWS1 interacts with casein kinase 1, and we show that this association is critical for PAWS1 to impact Wnt signalling in cells and embryos.

## Results

### PAWS1 induces the formation of a secondary axis in *Xenopus* embryos

In an effort to explore the biological activity of PAWS1, we injected 500 pg of mRNA encoding *Xenopus* PAWS1 into the animal hemispheres of *Xenopus* embryos at the one-cell stage. Such embryos went on to display axial defects, including dorsalisation and the formation of partial secondary axes (Fig EV1A–C). To explore this phenomenon in more detail, we injected a single ventral blastomere at the four-cell stage with xPAWS1 mRNA. Such embryos went on to form complete secondary axes, resembling those formed in response to ectopic xWnt8 (Fig 1A and B). Similar results were obtained with human PAWS1 (hPAWS1; Fig 1C).

1 Medical Research Council Protein Phosphorylation and Ubiquitylation Unit, Dundee, UK
2 The Francis Crick Institute, London, UK
*Corresponding author. Tel: +44 20 3796 1103; E-mail: jim.smith@crick.ac.uk
**Corresponding author. Tel: +44 1382 386330; E-mail: g.sapkota@dundee.ac.uk
†These authors contributed equally to this work

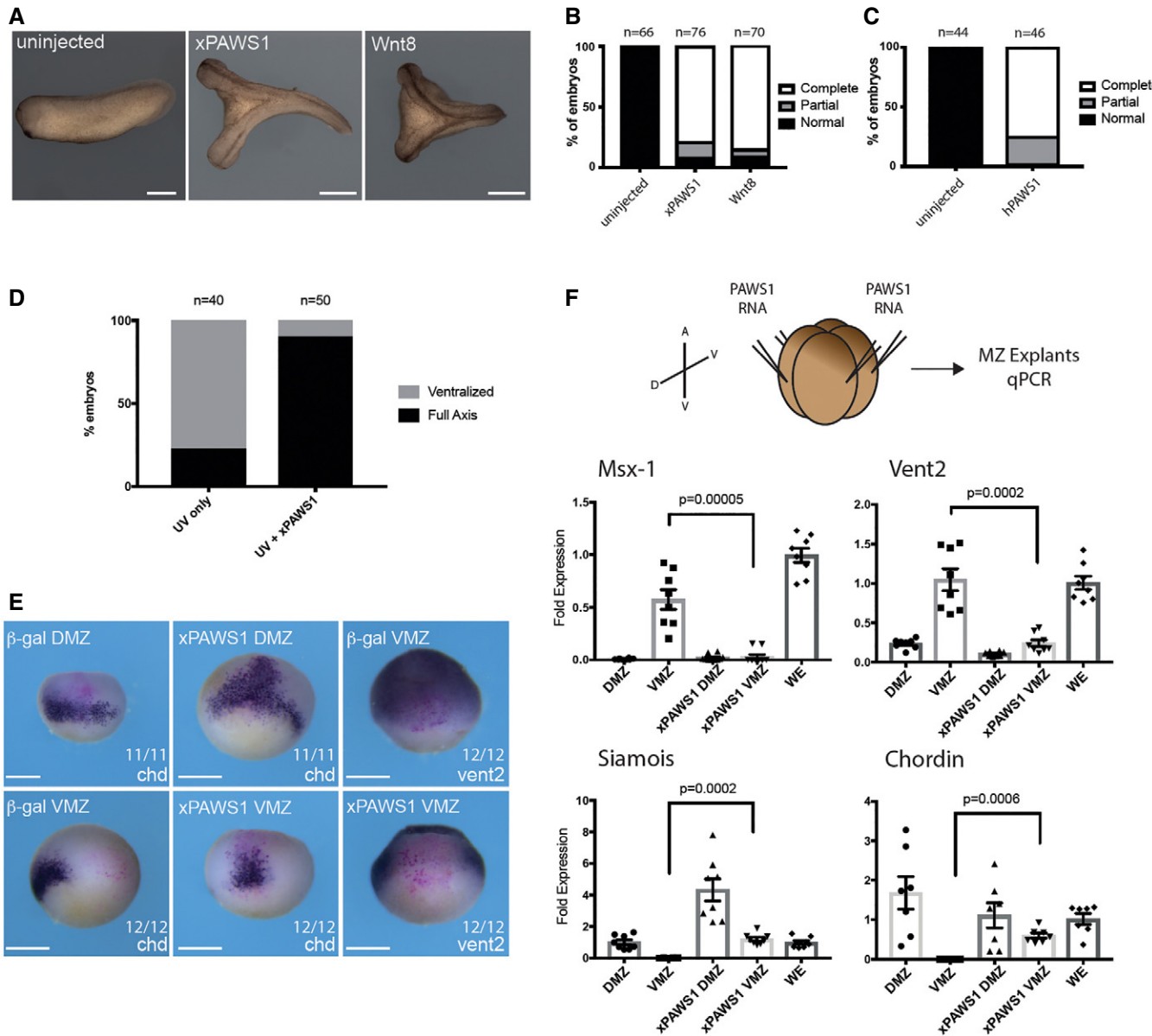

**Figure 1.  PAWS1 causes axis duplication in *Xenopus* embryos.**

A   Representative images of an uninjected embryo, and embryos injected with 250 pg xPAWS1 mRNA or 5 pg xWnt8 mRNA. Scale bars are 1 mm.

B   Quantification of (A). Complete axis denotes embryos with a secondary axis with a cement gland, while a partial axis does not.

C   Quantification of axis-inducing activity of 250 pg of hPAWS1.

D   250 pg of xPAWS1 mRNA rescues UV-ventralised embryos.

E   xPAWS1 induces dorsal and represses ventral gene expression in the whole embryo. A single blastomere at the four-cell stage was injected with either 200 pg nuclear β-gal mRNA alone or with 250 pg of xPAWS1 and 200 pg of nuclear β-gal mRNA. At stage 10, embryos were fixed, stained for β-gal and then developed for *in situ* hybridisation using probes specific for Chordin or Vent2. In dorsal blastomeres, xPAWS1 induces expression of Chordin, while in ventral blastomeres it represses Vent2 expression. The number of embryos tested is indicated. Scale bars are 500 μm.

F   250 pg of xPAWS1 and 200 pg of CFP_gpi mRNAs was injected into the marginal zone of each blastomere at the four-cell stage. The dorsal/ventral (D/V) and animal/vegetal (A/V) axes are indicated. At stage 10, the dorsal marginal zone (DMZ) and ventral marginal zone (VMZ) were isolated and dorsal and ventral marker expression was assessed by qPCR. xPAWS1 inhibits ventral marker expression (*Msx-1* and *Vent2*), while concomitantly upregulating the dorsal markers *Chordin* and *Siamois* in the VMZ. Expressions of *Msx-1*, *Vent2*, *Siamois*, and *Chordin* were normalised to the expression levels of *Histone H4*, and presented as a fold change with respect to the average levels in whole embryos (WE), ($n = 8$, error bars represent $\pm$ SEM; *t*-test, unpaired, two-tailed with unequal variance Mann–Whitney test, *P*-values are as indicated).

Early UV exposure of *Xenopus* embryos inhibits cortical rotation and prevents formation of a dorsal inducing centre [9]. UV-ventralised embryos of this sort can be rescued by ectopic activation of the Wnt signalling pathway [10–13], and consistent with its strong dorsalising activity, we found that xPAWS1 is indeed able to rescue UV-treated embryos (Fig 1D).

We also examined gene expression in the early embryo following xPAWS1 overexpression. When injected into the marginal zone, xPAWS1 suppressed expression of the ventral genes *Msx1* (Fig 1F), and *Vent2* in the ventral marginal zone (VMZ; Fig 1E and F), and increased expression of dorsal markers *Siamois* (Fig 1F), and *Chordin* (Fig 1E and F). Together, these results suggest that PAWS1 promotes Wnt signalling.

### PAWS1 does not inhibit canonical BMP signalling

Components of the Wnt signal transduction pathway can induce the formation of a complete secondary axis in *Xenopus* [14–17]. However, inhibition of BMP signalling can induce partial secondary axes [18], and it is possible that as a SMAD1-interacting protein, PAWS1 might also exert some effects by inhibiting BMP signalling. However, xPAWS1 did not inhibit BMP-induced nuclear translocation of SMAD1 in dissociated animal cap cells, nor did it affect expression of the BMP responsive ventral genes *Msx1* and *Vent1* (Fig 2A and B). Together, these data suggest that the ability of PAWS1 to induce secondary axes occurs solely through the Wnt pathway.

Consistent with this conclusion, we showed previously that the introduction of PAWS1 in PC3 prostate cancer cells (that normally do not express PAWS1) does not affect BMP-induced phosphorylation of SMAD1, nor the transcription of SMAD4-dependent BMP target genes [1]. Similarly, BMP-induced phosphorylation of SMAD1 in PAWS1$^{-/-}$ U2OS cells created by CRISPR/Cas9 genome editing [19] resembles that in wild-type U2OS cells as well as that in PAWS1$^{-/-}$ cells in which PAWS1 has been restored (Fig 2C). The same is true for TGFβ-induced phosphorylation of SMAD2/3 in the three cell lines (Fig 2C). Thus, PAWS1 has no significant effect on canonical BMP signalling in *Xenopus* embryos and nor in human cells.

### PAWS1 enhances canonical Wnt signalling in *Xenopus* embryos and human cells

Our data suggest that PAWS1 activates Wnt signalling. To confirm this, we tested its ability to stabilise GFP-tagged β-catenin in *Xenopus* animal pole regions. GFP fluorescence was undetectable in animal pole cells derived from embryos injected with 50 pg mRNA encoding β-catenin-GFP, either by Western blot (Fig 3A and C) or by confocal imaging (Figs 3E and EV1D, F and G). Some stabilised β-catenin-GFP was detectable after treatment of caps with the Wnt agonist CHIR99021 (Fig EV1E, H and I). Approximately ten times more β-catenin-GFP was present in animal caps co-expressing xPAWS1 (Fig 3A and C), some of which was in an active unphosphorylated state (Fig 3A and C). β-Catenin-GFP could also be detected by confocal microscopy in the nuclei of dissociated animal cap cells (Fig 3E). Levels of endogenous and active β-catenin also

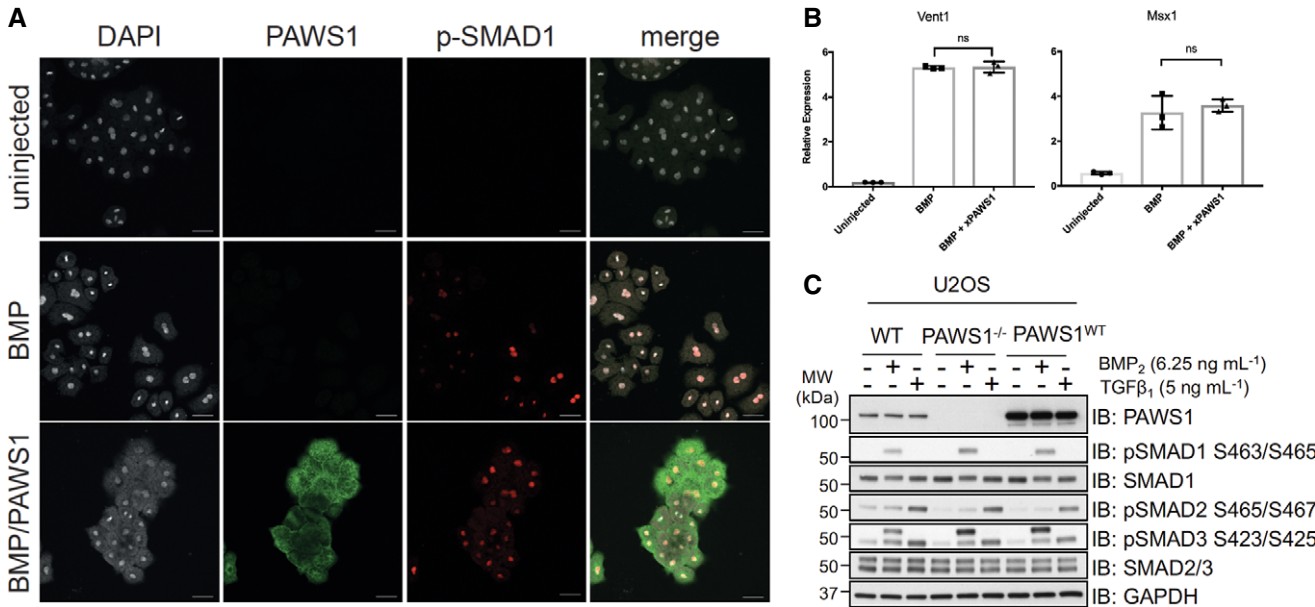

**Figure 2.  BMP signalling pathway is not affected by PAWS1.**

A   Expression of pSMAD1 in dissociated animals cap cells. Embryos were injected into the animal pole of both blastomeres at the two-cell stage with a total of either 1 ng of BMP4 or 1 ng BMP4 and 500 pg of Myc-tagged(MT)xPAWS1 mRNAs. Dissociated cells were stained with antibodies against MYC-tag (xPAWS1, green) and for phospho-SMAD1 (p-SMAD1, red). Scale bars are 50 μm.

B   Expression of ventral markers in animal caps cells injected at the two-cell stage with a total of either 1 ng of BMP4 or 1 ng BMP4 and 500 pg of MTxPAWS1 mRNA (n = 3, error bars represent ± SD, ns—not significant, *t*-test, unpaired, two-tailed with unequal variance).

C   U2OS wild-type (WT), PAWS1$^{-/-}$ and PAWS1$^{WT}$ rescue cells were serum-deprived for 16 h. Cells were subsequently stimulated with either 6.25 ng/ml BMP2 or 50 pM TGFβ1 for 1 h prior to lysis. Cell extracts (15 μg) were resolved by SDS–PAGE and immunoblotted with the indicated antibodies. Note that the upper band in the P-SMAD3 blot is a result of the antibody cross-reacting with P-SMAD1.

Source data are available online for this figure.

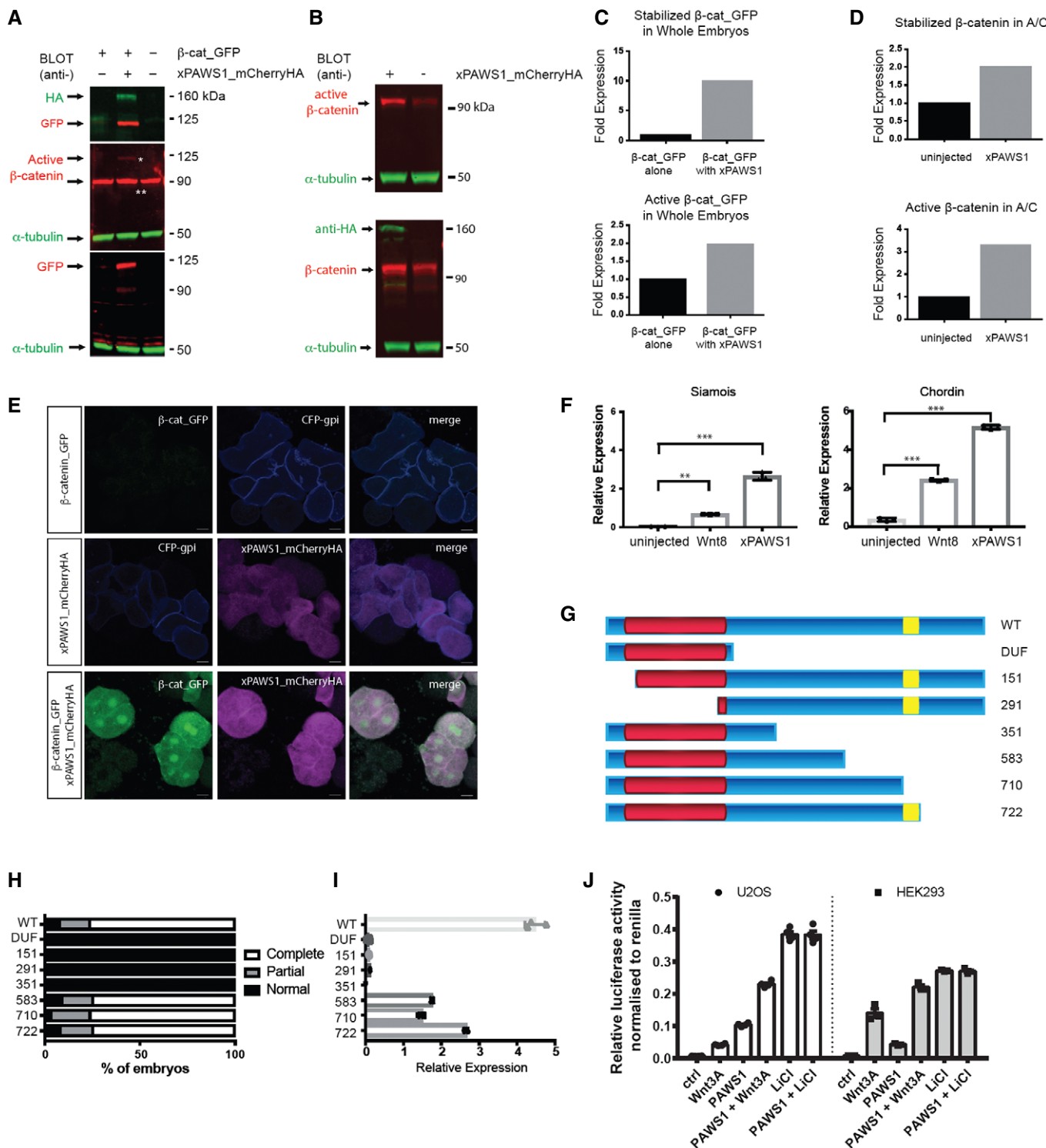

**Figure 3.**

increased following injection of xPAWS1 mRNA into naïve animal caps (Fig 3B and D). Finally, animal caps expressing xPAWS1 activated the direct Wnt target *Siamois* [20–22] and the dorsalising factor *Chordin* [18], confirming that PAWS1 can activate the canonical Wnt pathway (Fig 3F). We mapped the domain of PAWS1 that is required for Wnt signalling by testing the ability of different

PAWS1 fragments to induce secondary axes in *Xenopus* and to activate *Siamois* expression in animal caps (Fig 3G–I). The DUF1669 domain proved to be necessary but not sufficient to activate canonical Wnt signalling, while the 583 fragment that displays activity suggests that the conserved BMPR1 phosphorylation sites S[610], S[613] and S[614] do not impact activity (Fig 3G–I). The expression of these

**Figure 3.  PAWS1 activates Wnt signalling.**

A    xPAWS1 stabilises exogenous β-catenin in whole embryos. *Xenopus* embryos were injected into the animal pole at the one-cell stage with mRNAs encoding either 50 pg of β-cat_GFP or with 50 pg of β-cat_GFP and 250 pg xPAWS1_mCherryHA mRNAs. At stage 10, embryo extracts were immunoblotted with the indicated antibodies. Top panel: anti-HA and anti-GFP antibodies; middle panel: anti-active β-catenin (single white asterisk is active β-catenin_GFP; double white asterisk is endogenous active β-catenin) and anti-α-tubulin antibodies; bottom panel: anti-GFP and anti-α-tubulin antibodies.

B    xPAWS1 stabilises endogenous β-catenin in naïve animal caps. Embryos were injected at the one-cell stage with 250 pg of xPAWS1_mCherryHA mRNA. At stage 8.5, animal caps were collected from injected and uninjected embryos and cultured until control embryos reached stage 10. Extracts were immunoblotted with the indicated antibodies. Top panel: anti-active β-catenin and anti-α-tubulin antibodies; bottom panel: anti-HA, anti-β-catenin and anti-α-tubulin antibodies.

C, D    Quantification of (A and B), respectively. In (C), β-catenin_GPF or active β-catenin_GFP bands were normalised to α-tubulin and then expressed as a fold change relative to the expression of β-catenin_GFP and active β-catenin_GFP respectively from embryos injected with β-catenin_GFP alone. In (D), endogenous β-catenin and active β-catenin were normalised to α-tubulin and then expressed as a fold change relative to the expression of β-catenin and active β-catenin (respectively) from uninjected cells.

E    Nuclear translocation of β-catenin_GFP. Dissociated animal cap cells injected with either 50 pg of β-catenin_GFP or with 50 pg β-catenin_GFP and 250 pg xPAWS1_mCherryHA mRNAs were plated on coverslips and imaged by confocal microscopy. Only β-catenin_GFP cells co-injected with xPAWS1_mCherryHA mRNA accumulated robust levels of β-catenin in the nucleus. Scale bars are 20 μm.

F    xPAWS1 induces expression of *Siamois* and *Chordin* transcripts in animal caps. Embryos were injected at the one-cell stage with 250 pg of MT_xPAWS1 mRNA, and then at stage 8.5, animal caps were collected from injected and uninjected embryos and assessed for *Chordin* and *Siamois* expression by qPCR ($n = 3$; error bars represent $\pm$ SD, **$P = 0.001$, ***$P = 0.0001$; ordinary one-way ANOVA with multiple comparisons).

G–I    The DUF1669 domain (G, in red) is necessary but not sufficient to induce a secondary axis (H) and activate *Siamois* expression (I) while the BMPR1 phosphorylation sites $S^{610}$, $S^{613}$ and $S^{614}$ (yellow) are dispensable ($n = 3$; error bars represent $\pm$ SD). 250 pg of MT_xPAWS1 mRNAs encoding N- and C-terminal truncation fragments were injected into one ventral blastomere at the four-cell stage. Axis induction was assessed at stage 28. In (I), embryos were injected at the one-cell stage with 250 pg of MT_xPAWS1 mRNAs encoding N- and C-terminal mutants, and then at stage 8.5, animal caps were collected and assessed for Siamois expression by qPCR.

J    HEK293 and U2OS cells were transfected with PAWS1 cDNA, or empty vector as a control and TOPFlash luciferase activity was measured after treatment with either conditioned medium (L-CM), Wnt3A-conditioned medium (L3-CM) or 20 mM LiCl for 12 h. Data are normalised to Renilla internal control ($n = 4$; error bars represent $\pm$ SEM).

Source data are available online for this figure.

fragments in embryos was monitored by Western blotting (Fig EV1J).

We also asked whether PAWS1 activates Wnt signalling in mammalian cells. To this end, we overexpressed PAWS1 in HEK293 human kidney cells and in U2OS osteosarcoma cells before stimulating with control or Wnt3A-conditioned medium and measuring Wnt-dependent TOPFlash luciferase reporter activity [23]. In both cell lines, Wnt3A conditioned medium induced TOPFlash luciferase activity when compared with the control (Fig 3J). Overexpression of PAWS1 substantially enhanced activity under both conditions, as did inhibition of GSK-3 activity by treatment with LiCl (Fig 3J). Consistent with these observations, Wnt-induced TOPFlash luciferase activity in PAWS1$^{-/-}$ U2OS cells was significantly less than in wild-type controls (Fig 4A), as was Wnt-induced activation of its endogenous target genes *AXIN2* and *CYCLIN D1* (Fig 4B).

To ask where within the Wnt signalling pathway PAWS1 is likely to act, we employed epistasis experiments in animal caps. The induction of *Siamois*, a direct Wnt target, in naïve animal caps injected with xWnt8 mRNA is inhibited following injection of a dominant negative Wnt receptor Lrp6 m5 (Fig 4C). In contrast, Lrp6 m5 was unable to inhibit *Siamois* expression in caps injected with xPAWS1 mRNA, suggesting that PAWS1 acts downstream of the Wnt receptor (Fig 4D). Overexpression of C-cadherin in animal caps can inhibit Wnt signalling by acting as a molecular sink and sequestering stabilised β-catenin to the cell surface, thereby inhibiting nuclear accumulation of β-catenin [24]. Therefore, agonists that promote Wnt signalling downstream of β-catenin are unaffected by C-cadherin overexpression. PAWS1-dependent induction of Siamois expression was blocked by C-cadherin (Fig 4D), suggesting that PAWS1-mediated activation of Wnt signalling requires β-catenin. Ubiquitin-mediated degradation of β-catenin is regulated through the destruction complex, in which CK1, GSK-3 and Axin1 play key

roles [25–27]. Overexpression of Axin1 in animal caps completely blocked xPAWS1-induced Siamois expression, while overexpression of GSK-3 did so partially (Fig 4D). Consistent with this notion, in PAWS1$^{-/-}$ U2OS cells, inhibition of GSK-3 with the selective inhibitor CHIR99021 had no effect on Wnt-reporter activity (Fig 4E), while in WT cells inhibition of GSK-3, which stabilises β-catenin [28], yielded maximal induction of Wnt-reporter activity, even in the absence of Wnt stimulus. Together, these results suggest that PAWS1 functions in Wnt signalling at the level of the destruction complex or subsequent to it.

**PAWS1 interacts with CK1α**

To understand the mechanisms by which PAWS1 stimulates canonical Wnt signalling at the level of the destruction complex, we employed a proteomic approach to identify interactors of endogenous PAWS1. For this, we first generated U2OS cells in which we introduced a GFP tag at the C-terminus of the endogenous PAWS1 gene on both alleles (Fig EV1K and L). Using these PAWS1$^{GFP/GFP}$-knock-in cells, we performed anti-GFP IPs to pull down endogenous PAWS1-GFP and its interacting partners (Fig 5A). Casein kinase 1α (CK1α) isoform was identified as the major interactor of PAWS1 by mass spectrometry (Figs 5A and EV1M and N). CK1α plays a central role in Wnt signalling and phosphorylates many components of the pathway, including β-catenin at the destruction complex [29,30]. By Western blotting, endogenous CK1α was detected in anti-GFP IPs from PAWS1$^{GFP/GFP}$ knock-in cells but not from PAWS1$^{-/-}$ U2OS cells (Fig 5B). Similarly, endogenous PAWS1 was detected in endogenous CK1α IPs from wild-type but not PAWS1$^{-/-}$ U2OS cells (Fig 5C). Stimulating cells with Wnt3A did not affect the ability of PAWS1 to interact with endogenous CK1α (Fig EV1O), suggesting that the interaction between PAWS1 and CK1α does not depend on Wnt stimulation.

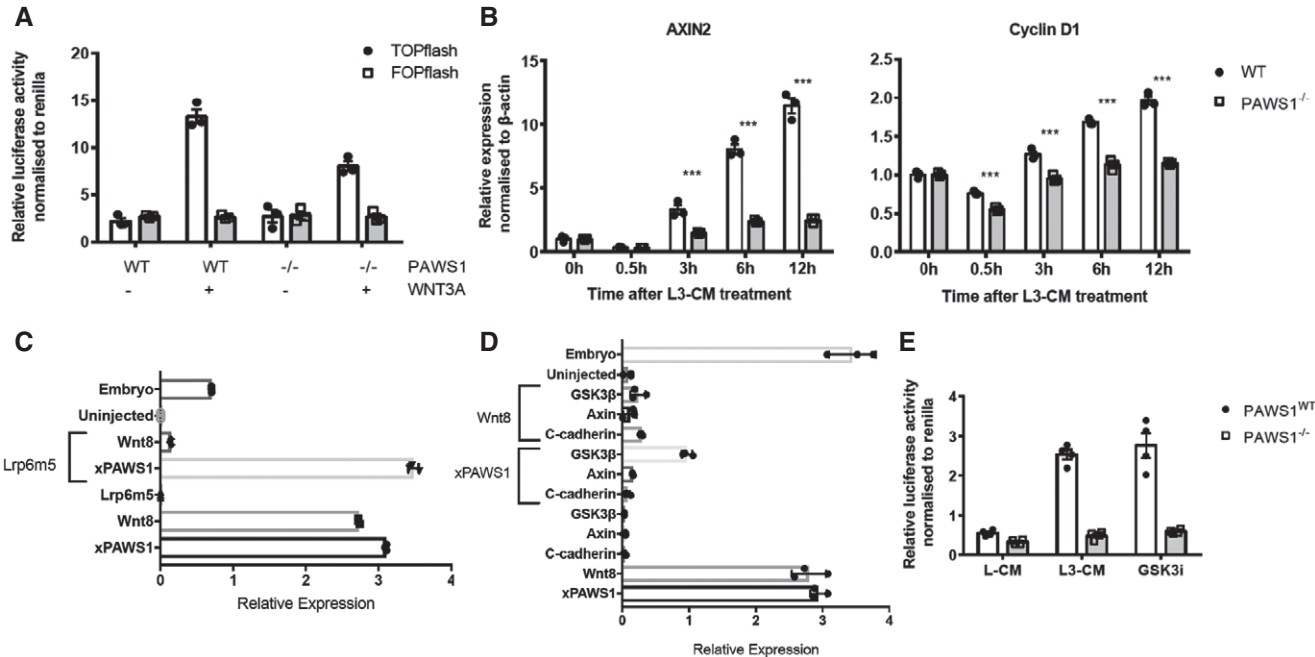

**Figure 4. Loss of PAWS1 inhibits Wnt signalling at the level of the destruction complex.**

A    Relative TOPFlash luciferase activity of PAWS1$^{WT}$ and PAWS1$^{-/-}$ U2OS cells after treatment with conditioned media (L-CM) or Wnt3A-conditioned media (L3-CM) ($n$ = 3; error bars represent ± SEM).

B    Wnt3A-induced activation of the target genes AXIN2 and CYCLIN D1 was examined by qPCR at the indicated time points in PAWS1$^{WT}$ and PAWS1$^{-/-}$ U2OS cells. Transcript expression data are represented as fold induction over unstimulated control and are internally normalised to β-actin control. Error bars represent ± SEM (***$P$ < 0.0001; two-way ANOVA with multiple comparisons; $n$ = 3).

C, D    Epistasis analysis of xPAWS1 in the canonical Wnt pathway. *Xenopus* embryos were injected at the one-cell stage with the indicated mRNAs, and then at stage 10, animal caps were assessed by qPCR for the expression of Siamois. xPAWS1 acts downstream of the Wnt receptor LRP6 (C) at the level of the destruction complex (D) ($n$ = 3 error bars represent ± SD).

E    Relative TOPFlash luciferase activity of PAWS1$^{WT}$ and PAWS1$^{-/-}$ U2OS cells after treatment with either conditioned medium (L-CM), Wnt3A-conditioned medium (L3-CM) or 5 μM of the GSK3 inhibitor CHIR99021 for 6 h. Data are normalised to Renilla-luciferase internal control ($n$ = 4; error bars represent ± SEM).

To map the interaction sites between PAWS1 and CK1α, we co-expressed truncated fragments of PAWS1 with CK1α and performed co-IP interaction assays. From these assays, we determined that the region spanning residues 165-307 within the DUF1669 domain was sufficient for mediating CK1α interaction [31]. Within this region, PAWS1 contains an F$^{296}$-X-X-X-F$^{300}$ motif that has been reported to mediate interaction between CK1α and NFAT1, PER1 and PER2 proteins [32]. In order to determine whether the F$^{296}$-X-X-X-F$^{300}$ motif within PAWS1 indeed mediates its association with CK1α, we generated FLAG-PAWS1 F296A, F300A or F296A/F300A mutants and overexpressed them in U2OS cells. Endogenous CK1α was detected in anti-FLAG IPs from wild-type FLAG-PAWS1 and FLAG-PAWS1$^{F300A}$ but not from FLAG-PAWS1$^{F296A}$ and FLAG-PAWS1$^{F296A/F300A}$ (Fig 5D), suggesting F$^{296}$ in PAWS1 contributed to the association with CK1α. For PER1/2, it has been shown that both Phe residues within the F-X-X-X-F motif are essential for interaction with CK1α, as mutating either contributed to the loss of interaction with CK1 [32]. However, for PAWS1, the fact that PAWS1$^{F300A}$ was still able to interact with CK1α suggested that the mode of interaction with CK1α is mediated through additional mechanisms. We tested further point mutations on conserved residues along the aa165-aa307 stretch of PAWS1 for their ability to abolish the interaction with CK1α and found that mutating the

conserved D$^{262}$ to A (PAWS1$^{D262A}$) also abolished the association with CK1α (Fig 5D). We tested the ability of GFP control, PAWS1$^{WT}$, PAWS1$^{F296A}$ or PAWS1$^{D262A}$ to interact with endogenous CK1α when these were stably restored in PAWS1$^{-/-}$ cells. IPs of endogenous CK1α pulled down PAWS1 only from PAWS1$^{WT}$ rescue cells but not others (Fig 5E).

To test whether, and if so where, PAWS1 and CK1α co-localise in cells, we restored the expression of PAWS1$^{WT}$ or PAWS1$^{F296A}$ in PAWS1$^{-/-}$ U2OS cells. By immunofluorescence, we observed that PAWS1$^{WT}$ and CK1α displayed overlapping cytoplasmic staining, while PAWS1$^{F296A}$ and CK1α displayed non-overlapping cytoplasmic staining and, as expected, no PAWS1 was detected in PAWS1$^{-/-}$ cells (Fig 5F). Interestingly, the CK1α staining in PAWS1$^{-/-}$ and PAWS1$^{F296A}$ rescue cells was much weaker than in PAWS1$^{WT}$ rescue cells (Fig 5F), suggesting that PAWS1 might play a role in cellular CK1α protein levels.

## PAWS1 regulates cellular CK1α protein levels

The robust association and co-localisation of PAWS1 with CK1α has several implications for the role of PAWS1 in the Wnt signalling pathway. These include (i) PAWS1 is a substrate of CK1α; (ii) PAWS1 controls CK1α activity; and (iii) PAWS1 controls CK1α

    

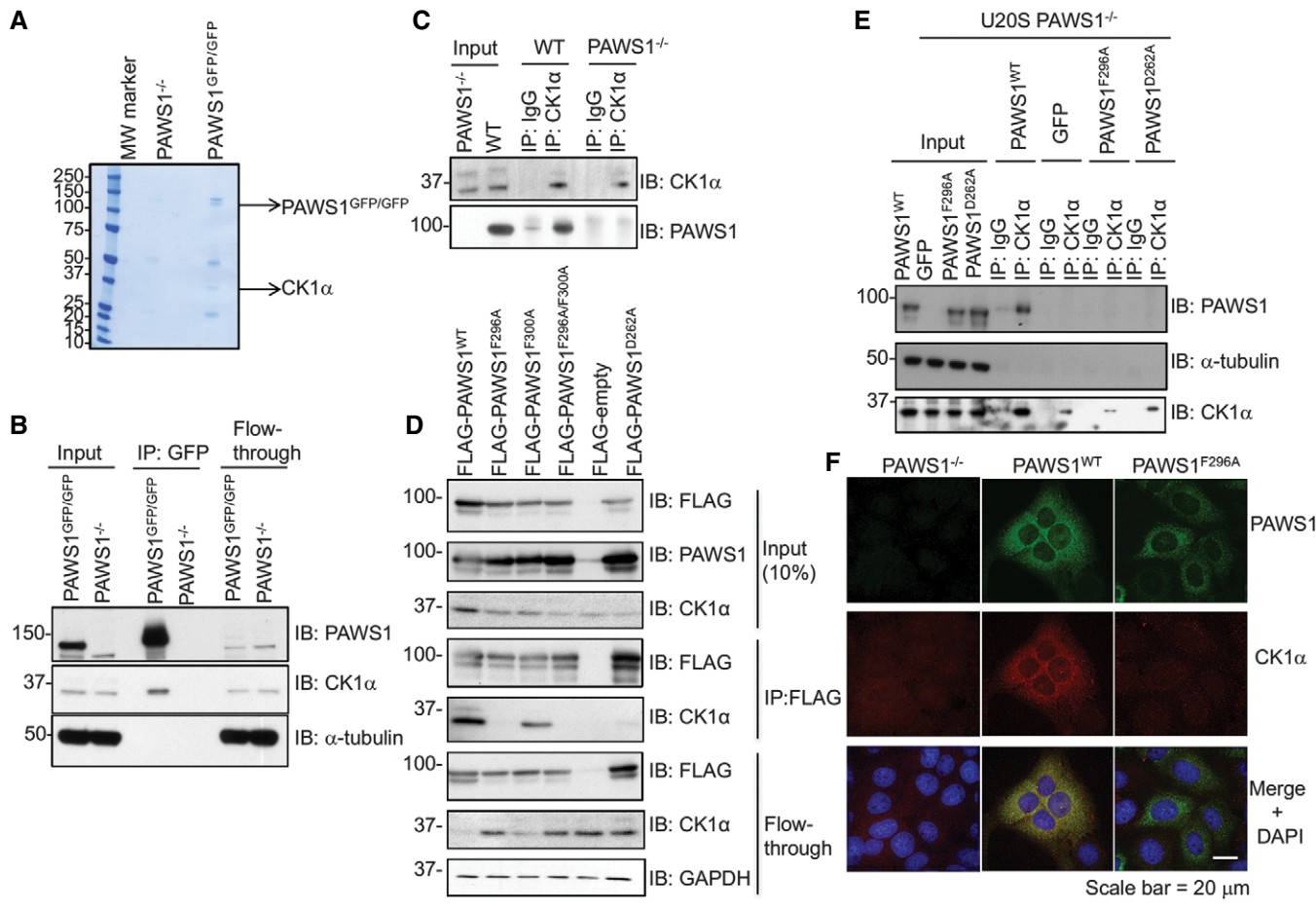

**Figure 5.  PAWS1 interacts with CK1 at the endogenous level.**

A  GFP pull downs from U2OS PAWS1$^{-/-}$ and PAWS1$^{GFP/GFP}$ cells were resolved by SDS–PAGE, and the gel was stained with Coomassie. Each lane was cut into six pieces, which were subsequently processed for protein identification by mass spectrometry.

B  GFP pull downs from U2OS PAWS1$^{-/-}$ and PAWS1$^{GFP/GFP}$ cells were resolved by SDS–PAGE and analysed by Western blotting using the indicated antibodies.

C  Anti-CK1α IPs from U2OS PAWS1$^{WT}$ and PAWS1$^{-/-}$ cells were resolved by SDS–PAGE and analysed by Western blotting using the indicated antibodies.

D  U2OS cells were transiently transfected with cDNA encoding FLAG-tagged PAWS1$^{WT}$, PAWS1$^{F296A}$, PAWS1$^{F300A}$, PAWS1$^{F296A/F300A}$, PAWS1$^{D262A}$ and FLAG-empty vector. Anti-FLAG IPs were immunoblotted with the indicated antibodies.

E  Anti-CK1α IPs and IgG-IPs from U2OS PAWS1$^{WT}$, GFP, PAWS1$^{F296A}$ and PAWS1$^{D262A}$ rescue cells were resolved by SDS–PAGE and analysed by Western blotting using the indicated antibodies.

F  PAWS1 co-localises with CK1α in U2OS cells. PAWS1$^{-/-}$ cells in which PAWS1$^{WT}$ or PAWS1$^{F296A}$ expression was restored were fixed for immunofluorescence using antibodies against PAWS1 and CK1α. Scale bar is 20 μm. Images from one field of view representative of three biological replicates are included.

Source data are available online for this figure.

stability and/or levels in cells. We sought to test these possibilities.

*In vitro*, CK1α phosphorylated PAWS1 and we mapped the CK1α phosphorylation site on PAWS1 to Ser$^{614}$ using Edman sequencing and mass spectrometry (Fig EV2A–C). Consistent with this, CK1α was no longer able to phosphorylate PAWS1$^{S614A}$ mutant (Fig EV2D). However, bearing in mind that the PAWS1(1-583) fragment, which lacks Ser$^{614}$, was still able to cause axis duplication and activate Wnt signalling in *Xenopus* embryos (Fig 3G–I), the phosphorylation of PAWS1 at Ser$^{614}$ by CK1α is unlikely to explain its role in Wnt signalling. Furthermore, the PAWS1$^{S610A/S614A}$ mutant was still able to induce axis duplication (Fig EV2E). As yet, we have not determined whether endogenous PAWS1 is phosphorylated at Ser$^{614}$ by CK1α *in vivo*.

We also tested the ability of PAWS1 to influence CK1α activity against a peptide substrate (CK1tide) [33]. First, we detected robust CK1α kinase activity in endogenous PAWS1 IPs from wild-type but not PAWS1$^{-/-}$ U2OS cell extracts (Fig 6A). The detection by Western blotting of endogenous CK1α in PAWS1 IPs from wild-type but not PAWS1$^{-/-}$ cell extracts correlated with the observed *in vitro* kinase activities against CK1tide (Fig 6A). These observations suggest that the association of PAWS1 with CK1α does not inhibit intrinsic CK1α kinase activity. Next, we tested whether PAWS1 potentiated the kinase activity of recombinant CK1α through association. Human recombinant CK1α expressed in *E. coli* displayed *in vitro* kinase activity against CK1tide, while the catalytically inactive (KD; D136N) mutant of CK1α did not (Fig 6B). The addition of recombinant human PAWS1$^{WT}$ or the CK1-interaction-deficient

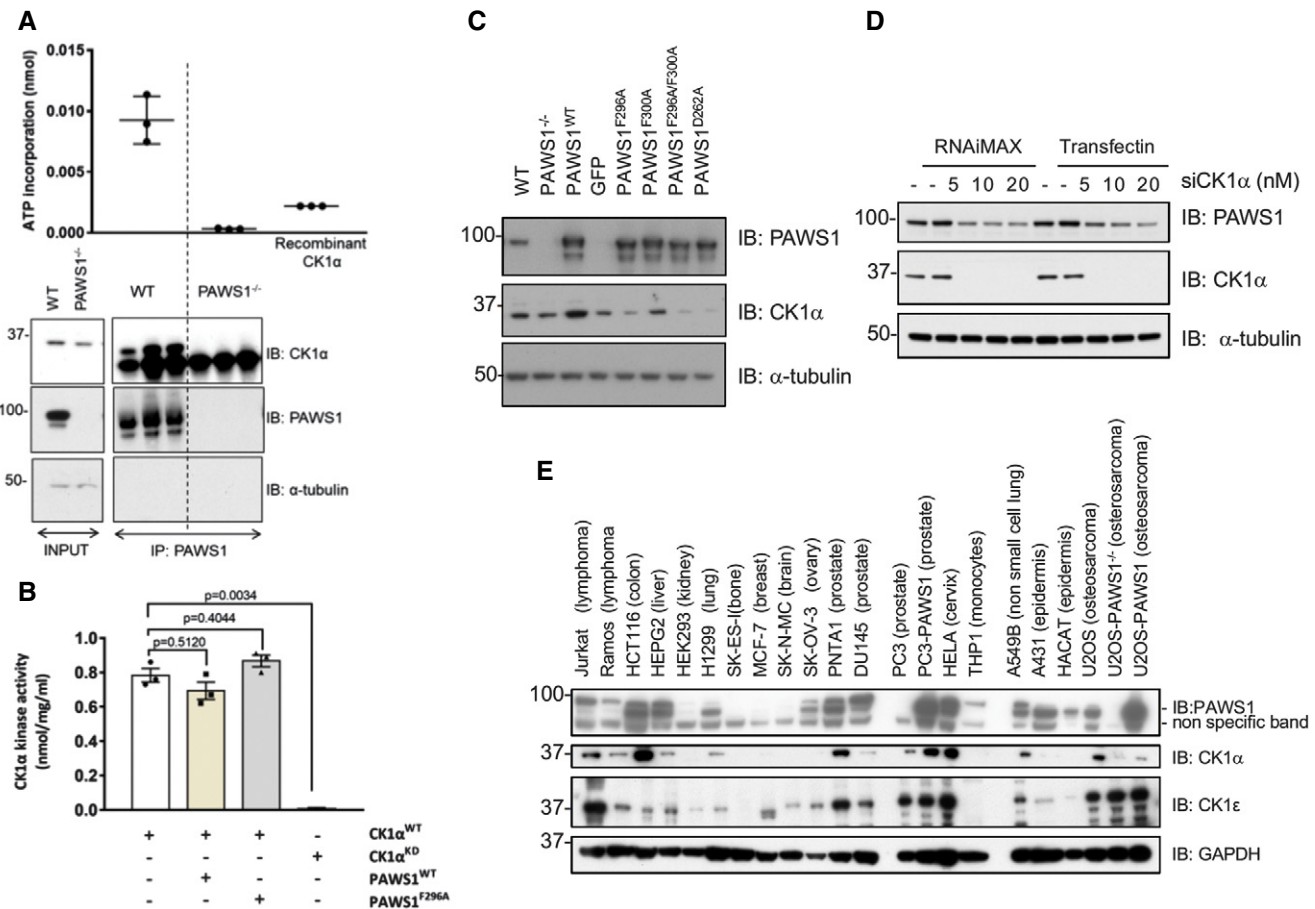

**Figure 6. PAWS1 does not affect CK1α activity but affects the protein levels of CK1α in cells.**

A   Endogenous PAWS1 was immunoprecipitated from wild-type and PAWS1$^{-/-}$ U2OS cells ($n$ = 3), and the associated CK1α kinase activity was measured following a kinase assay using $^{\gamma 32}$P-ATP and the CK1tide peptide substrate. A third of the PAWS1 IP samples were resolved by SDS–PAGE and immunoblotted with the indicated antibodies (bottom panel) ($n$ = 3, error bars represent ± SEM).

B   *In vitro* wild-type (WT) or catalytically inactive (KD) CK1α kinase assay using CK1tide as a substrate in the presence or absence of recombinant PAWS1$^{WT}$ or PAWS1$^{F296A}$ ($n$ = 3; error bars represent ± SEM; one way ANOVA with multiple comparisons, n.s.: no statistical significance).

C   Cell extracts from U2OS wild-type, PAWS1$^{-/-}$ and PAWS1$^{-/-}$ cells in which PAWS1, GFP control, PAWS1$^{F296A}$, PAWS1$^{F300A}$, PAWS1$^{F296A/F300A}$ or PAWS1$^{D262A}$ was restored, were resolved by SDS–PAGE and immunoblotted with the indicated antibodies.

D   CK1α expression was silenced in U2OS cells with indicated amounts of siRNA using two different transfection reagents (RNAi-Max or TransFectin) and PAWS1 protein levels monitored by Western blotting.

E   Expression of PAWS1 and CK1α protein in the indicated cell lines was monitored by Western blotting. The protein levels of CK1α mirror the levels of PAWS1 in the majority of the cancer cell lines tested.

Source data are available online for this figure.

PAWS1$^{F296A}$ mutant to the kinase reaction did not significantly affect the ability of CK1α to phosphorylate CK1tide (Fig 6B). Collectively, these data suggest that PAWS1 is unlikely to directly regulate the intrinsic CK1α kinase activity.

Strikingly, when we analysed the levels of CK1α protein in extracts from wild-type and PAWS1$^{-/-}$ U2OS cells, we observed decreased CK1α protein levels in PAWS1$^{-/-}$ cells compared to the WT controls (Fig 6C). When we rescued PAWS1$^{WT}$ in PAWS1$^{-/-}$ cells, we noticed a remarkable increase in the endogenous CK1α protein levels compared to the re-introduction of GFP control (Fig 6C). Interestingly, the re-introduction of the CK1α-binding-deficient mutants PAWS1$^{F296A}$ and PAWS1$^{F296A/F300A}$ did not result in any increased CK1α protein levels, while the PAWS1$^{F300A}$ mutant, which interacts with CK1α, resulted in increased CK1α

protein levels similar to PAWS1$^{WT}$ rescue (Fig 6C), suggesting that the binding of PAWS1 to CK1α regulates CK1α protein levels in cells. The reduction in levels of CK1α protein in PAWS1$^{-/-}$ U2OS cells compared to the WT controls could not be rescued by either the proteasomal inhibitor Bortezomib or the lysosomal inhibitor Bafilomycin A1 (Fig EV3A and B). Similarly, the mRNA levels of CK1α were comparable in PAWS1$^{WT}$, PAWS1$^{-/-}$ and PAWS1$^{F296A}$ rescue cells (Fig EV3E), suggesting that the PAWS1 regulation of CK1α protein levels is not through its transcriptional regulation. In U2OS cells in which we transiently depleted endogenous CK1α protein with siRNAs, we observed a concomitant reduction in endogenous PAWS1 protein levels (Fig 6D), suggesting a reciprocal effect of PAWS1 and CK1α on controlling cellular protein levels.

Prompted by these observations, we examined the potential correlation in expression of PAWS1 and CK1α proteins in a number of mammalian cell lines. We observed that PAWS1 protein expression was absent in many human cancer cells, including SK-ES (bone), MCF-7 (breast), SK-N-MC (brain), and PC3 (prostate) cells while robust expression was observed in HCT116 (colorectal), HEPG2 (liver), PNTA1 (prostate) and HeLa (cervical) cells. Strikingly, the expression levels of PAWS1 in these cells correlated with the levels of CK1α protein (Figs 6E and EV3C). Cells that lacked PAWS1 protein expression generally correlated with no detectable or low levels of CK1α protein, whereas cells that had relatively higher levels of PAWS1 also expressed relatively higher levels of CK1α (Figs 6E and EV3C). No correlation in expression was evident between PAWS1 and CK1ε, which does not interact with PAWS1 (Figs 6E and EV3D). In both PC3 and PAWS1$^{-/-}$ U2OS cells, which lack endogenous PAWS1, the restoration of WT PAWS1 increased the protein levels of endogenous CK1α but not CK1ε (Fig 6E) [31]. In order to test whether the correlation in PAWS1 and CK1α protein levels was due to a similar correlation in mRNA expression, we assessed the PAWS1 and CK1α mRNA levels in six of these mammalian cell lines. No discernible correlation was observed with the expression of PAWS1 and CK1α mRNAs (Fig EV3F–H), indicating that the correlation between PAWS1 and CK1α protein levels occurs post-transcriptionally.

### Interaction between PAWS1 and CK1α is essential for PAWS1-dependent axis duplication and the activation of Wnt signalling

Armed with the knowledge that PAWS1 associates and co-localises with CK1α, we asked whether the association between PAWS1 and CK1α was essential for PAWS1-dependent activation of Wnt signalling in Xenopus embryos and in U2OS cells. PAWS1-induced axis duplication in Xenopus embryos was scored following microinjection with wild-type PAWS1, three separate CK1α-interaction-deficient PAWS1 mutants, PAWS1$^{D262A}$, PAWS1$^{F296A}$ and PAWS1$^{F296A/F300A}$ as well as PAWS1$^{F300A}$ mutant that interacts with CK1α (Fig 7A). Both PAWS1$^{WT}$ and the PAWS1$^{F300A}$ mutant induced partial or complete axis duplication in ~80% of embryos. In contrast, PAWS1$^{D262A}$, PAWS1$^{F296A}$ and PAWS1$^{F296A/F300A}$ failed to induce secondary axes in the embryos (Fig 7A). These results demonstrate that the interaction between PAWS1 and CK1α is essential for PAWS1-mediated axis duplication in Xenopus embryos. To explore whether the induction of the Wnt-reporter activity in U2OS cells also required interaction between PAWS1 and CK1α, we assessed the Wnt-reporter activity in wild-type, PAWS1$^{-/-}$ and PAWS1$^{-/-}$ cells rescued with either PAWS1$^{WT}$ or the PAWS1$^{F296A}$ mutant. The treatment of wild-type cells with Wnt3A substantially enhanced the luciferase reporter activity over unstimulated controls, while in PAWS1$^{-/-}$ cells it did not (Fig 7B). In PAWS1$^{-/-}$ cells rescued with wild-type PAWS1, the Wnt3A-induced luciferase reporter activity was restored. In contrast, it was not restored in PAWS1$^{-/-}$ cells rescued with the PAWS1$^{F296A}$ mutant (Fig 7B), suggesting that PAWS1 activation of Wnt signalling necessitates PAWS1: CK1α interaction.

### The role of PAWS1:CK1α complex in Wnt signalling

In the absence of Wnt ligands, CK1α has been reported to phosphorylate β-catenin on Ser$^{45}$ [34,35], which triggers a sequential phosphorylation by GSK-3 on Thr$^{41}$, Ser$^{37}$ and Ser$^{33}$ [28], thereby allowing recruitment of the E3 ubiquitin ligase β-TrCP to β-catenin for its ubiquitin-mediated proteasomal degradation [26]. Paradoxically, overexpression of CK1α has been reported to stimulate Wnt signalling [36], and indeed, human CK1α can also induce a secondary axis in a dose-dependent manner in Xenopus embryos (Fig 7C). However, when co-expressed with PAWS1, the axis induction was blocked (Fig 7D). A possible explanation is that PAWS1 inhibits CK1α activity; hence, the overexpression of CK1α counters the PAWS1 effects on axis duplication. However, we have shown earlier that PAWS1 IPs from WT but not PAWS1$^{-/-}$ U2OS cells display CK1α catalytic activity and recombinant PAWS1 does not impair CK1α activity in vitro (Fig 6A and B). Therefore, these observations suggest that overexpression of either PAWS1 or CK1α alone in embryos potentially disrupts the endogenous PAWS1:CK1α complex homeostasis, resulting in the interference with Wnt signalling that results in the axis phenotypes observed. In contrast, when these are co-expressed, this interference on the endogenous PAWS1:CK1α complexes is minimised, so the phenotypes are normal. If this were the case, then one would expect similar outcomes using a catalytically inactive mutant of CK1α (Fig 7E), which we have shown interacts with PAWS1 with similar affinity to the wild-type CK1α (Fig 7F). Indeed, co-expression of PAWS1 and the catalytically inactive CK1α mutant in Xenopus embryos also resulted in completely normal embryos (Fig 7E). In similar assays, we also tested the ability of CK1δ and CK1ε, which do not interact with PAWS1 [31], to inhibit PAWS1-dependent axis duplication phenotype in Xenopus embryos. Unlike CK1α, CK1δ and CK1ε or the catalytically inactive CK1ε were unable to block the axis duplication phenotype caused by PAWS1 (Fig 7E). Like CK1α, CK1δ and CK1ε isoforms on their own also induced axis duplication in Xenopus embryos (Fig 7E), suggesting PAWS1-independent roles in Wnt signalling. Interestingly, catalytically inactive CK1α and CK1ε on their own did not induce axis duplication, suggesting that the catalytic activity is critical to their impact on Wnt signalling. Together, our findings suggest that critical regulation of endogenous PAWS1:CK1α complex is vital in driving Wnt signalling through axis specification during development and subtle imbalances can interfere with normal development. Not surprisingly, this feature is shared with the non-canonical Wnt/planar cell polarity pathway in which slight perturbations lead to defects in developmental events such as convergent extension and ciliogenesis [37].

Since PAWS1 appears to exist predominantly in a PAWS1:CK1α complex (Fig 5), we asked whether PAWS1:CK1α also regulated other CK1α-dependent pathways. One potential target of PAWS1:CK1α is the transcription factor NFAT, which has been reported to promote ventral cell fates in the Xenopus embryo [38]. Nuclear localisation of NFAT is regulated by CK1α via a SSR-1 motif in the N-terminus, in which phosphorylation of the motif by CK1 promotes NFAT1 nuclear export, while its dephosphorylation by calcineurin induces nuclear import. Phospho-deficient SSR-1 motif mutants of NFAT1 accumulate in the nucleus [32], as does NFAT in cells treated with a CK1 inhibitor [32], and when overexpressed in Xenopus ventralise embryos [38]. In dissociated animal cap cells expressing a NFAT_GFP reporter construct, NFAT_GFP accumulated predominantly in the cytoplasm (Fig EV4A–C). However, unlike cells expressing the catalytically inactive CK1α (KD), in which NFAT was distributed in the both the cytoplasm and the nucleus (Fig EV4C), we found that xPAWS1 expression had no effect on the

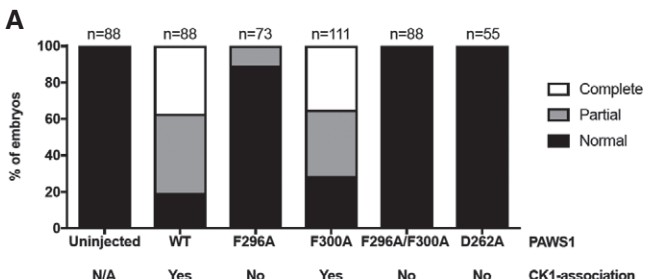

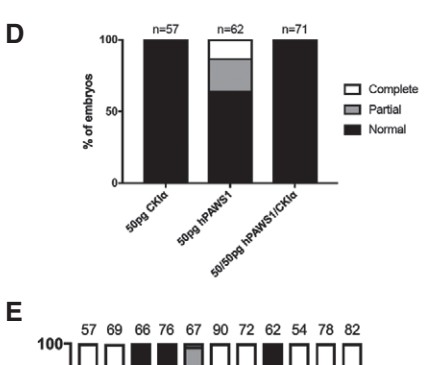

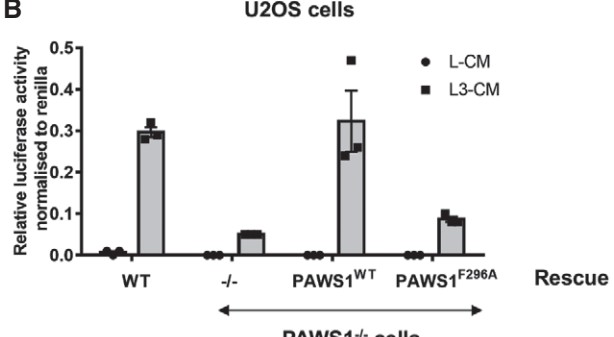

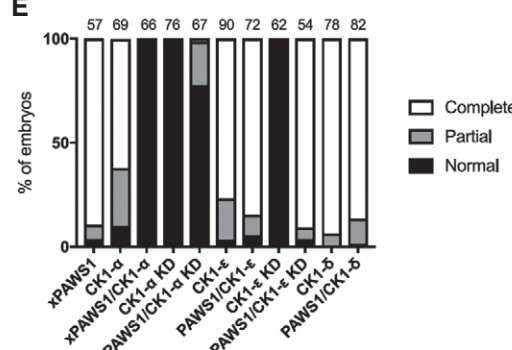

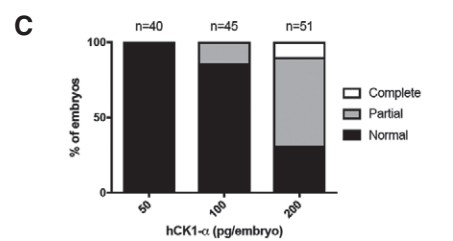

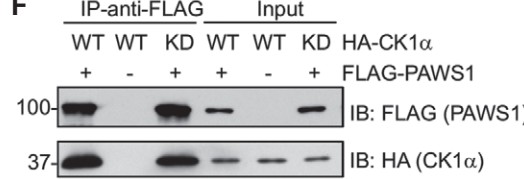

**Figure 7. PAWS1 association with CK1α is essential for induction of Wnt signalling.**

A   PAWS1$^{F296A}$, PAWS1$^{F296A/F300A}$ and PAWS1$^{D262A}$ mutants that do not interact with CK1α fail to induce a secondary axis in *Xenopus* embryos.

B   Relative TOPFlash luciferase activity of U2OS WT, PAWS1$^{-/-}$, PAWS1$^{WT}$ and PAWS1$^{F296A}$ rescue cells after treatment with control-conditioned media (L-CM), Wnt3A-conditioned media (L3-CM) (*n* = 3; error bars represent ± SEM).

C   Human CK1α induces a secondary axis in *Xenopus* embryos in a dose-dependent manner.

D   Co-expression of PAWS1 and CK1α mRNAs do not induce secondary axis.

E   Effect of CK1 isoforms on PAWS1-dependent axis-inducing activity. 250 pg of xPAWS1 and 300 pg of the indicated CK1 isoform mRNAs were injected into one ventral blastomere at the four-cell stage. Axis induction was assessed at stage 28.

F   PAWS1 interacts with kinase-dead CK1α in U2OS cells. U2OS cells were co-transfected with HA-CK1α (WT or KD) and FLAG-PAWS1, and cell extracts were subjected to FLAG IP, followed by SDS–PAGE and Western blot analysis with the indicated antibodies.

Source data are available online for this figure.

nuclear localisation of the reporter (Fig EV4A–C), nor did it affect nuclear localisation of NFAT_GFP following Ca²⁺ ionophore treatment (Fig EV4A–C) where no cytotoxicity was detected. These data suggest that, in contrast to the Wnt pathway, the regulation of nuclear localisation of NFAT is insensitive to perturbations of the PAWS1:CK1α complex.

**PAWS1 does not interact robustly with Axin1 and β-catenin complexes during Wnt signalling**

Having earlier established the role of PAWS1 in Wnt signalling at or downstream of the destruction complex, we sought to explore the composition of the destruction complex in the absence of PAWS1. In order to dissect any changes in the make-up of the destruction

complex, we immunoprecipitated CK1α, Axin1 and β-catenin from unstimulated and Wnt3A (3 h)-stimulated WT and PAWS1$^{-/-}$ U2OS cells and analysed some of the known components of the destruction complex. In endogenous CK1α IPs from both WT and PAWS1$^{-/-}$ cells, Axin1, GSK-3α/β and β-catenin were not detectable, regardless of Wnt3A stimulation, while as expected PAWS1 was identified in CK1α IPs from WT cells but not PAWS1$^{-/-}$ cells (Fig 8). In Axin1 IPs, GSK-3α/β were detected robustly, while very low levels of β-catenin were also detected, although neither Wnt3A treatment nor the PAWS1-status had any effect on these Axin1 complexes (Fig 8). Similarly, in β-catenin IPs, we were able to detect low levels of GSK-3β, as well as CK1δ and CK1ε but not CK1α (Fig 8), suggesting that CK1δ and CK1ε, rather than CK1α, may be essential in mediating β-catenin phosphorylation at Ser$^{45}$. CK1α IPs

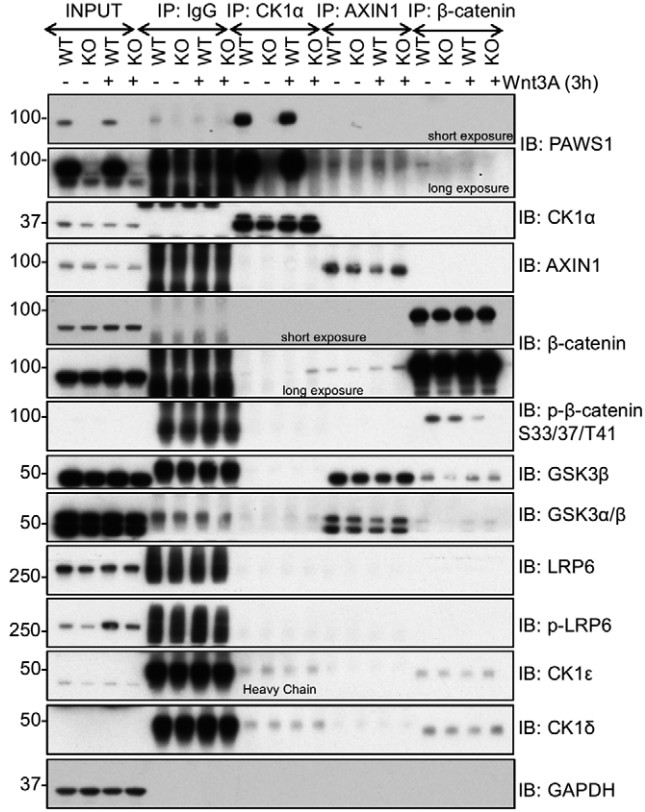

**Figure 8. PAWS1 does not appear to affect Axin1 and β-catenin complexes upon Wnt signalling.**

U2OS wild-type (WT) and PAWS1$^{-/-}$ (KO) cells were treated with control-conditioned medium or Wnt3A-conditioned medium, and the extracts (0.5 mg protein) were subjected to immunoprecipitation using antibodies against the endogenous CK1α, AXIN1 and β-catenin antibodies or anti-rabbit pre-immune IgG as a control (10 μg antibodies coupled to 10 μl packed protein-G sepharose beads). IPs were resolved by SDS–PAGE and immunoblotted with the indicated antibodies.

Source data are available online for this figure.

from cells treated with Wnt3A for shorter times (30 min and 1 h) did not pull down any significant levels of β-catenin and no differences were observed between WT and PAWS1$^{-/-}$ U2OS cells (Fig EV5A). With no significant differences in Axin1, GSK-3α/β and β-catenin complexes detected in the absence of PAWS1, we undertook an unbiased proteomic approach to identify interactors of PAWS1-GFP expressed in U2OS cells under unstimulated and Wnt3A (3 h)-stimulated conditions (Fig EV5B). However, other than CK1α and CK1α-like, no major components of either the Wnt pathway or the destruction complex were identified in the PAWS1-GFP IPs under both control and Wnt3A-stimulated conditions (Fig EV5C). While some PAWS1 interactors, such as PLOD1 and many 14-3-3 proteins, were substantially enriched in Wnt3A-stimulated conditions, albeit with much lower abundance compared to CK1α and CK1α-like, these still need to be validated further and their roles in Wnt signalling are not well established (Fig EV5B). As 14-3-3 proteins are binders of phospho-proteins [39], we analysed all phospho-residues on PAWS1 by mass spectrometry. Interestingly, some phospho-residues on PAWS1, namely Ser$^{127}$, Ser$^{634}$ and Ser$^{726}$, appeared to be enriched by Wnt3A stimulation (Fig EV5D).

Clearly, these need to be validated at the endogenous level and the functional relevance of these is unclear.

### PAWS1 impacts the nuclear translocation of β-catenin downstream of the destruction complex

Next, we assessed the effect of PAWS1-loss on the post-translational regulation of β-catenin. To this end, we analysed β-catenin phosphorylation on known CK1 (Ser$^{45}$) and GSK3 (Thr$^{41}$, Ser$^{37}$ and Ser$^{33}$) phospho-sites and its subcellular distribution. In WT U2OS cells, Wnt3A stimulation resulted in acute reduction in levels of both phospho-Ser$^{45}$ and phospho-Thr$^{41}$, -Ser$^{37}$ and -Ser$^{33}$ residues within 30 min, while the levels then recovered in a time-dependent manner (Fig 9A). This is in agreement with the accumulation of active non-phosphorylated (Thr$^{41}$, Ser$^{37}$ and Ser$^{33}$) form of either endogenous β-catenin or ectopic β-catenin-GFP observed in *Xenopus* animal caps injected with xPAWS1 (Fig 3A–D). No significant differences in total endogenous β-catenin levels were observed over the Wnt3A stimulation time course in either cell type, suggesting that in U2OS cells, only a small pool of total β-catenin might be regulated by Wnt3A treatment (Fig 9A). Interestingly, under unstimulated basal conditions in which cells were grown in DMEM containing 10% serum, slightly higher levels of both CK1α (Ser$^{45}$) and GSK3 (Thr$^{41}$, Ser$^{37}$ and Ser$^{33}$) phospho-sites in β-catenin were observed in PAWS1$^{-/-}$ extracts compared to the wild type (Fig 9A). No substantial differences between the wild-type and PAWS1$^{-/-}$ cells were observed when we analysed the levels of Wnt-induced phosphorylation of LRP6 at Ser$^{1490}$ (Fig 9A). We also assessed the expression of AXIN2 protein, a target of Wnt, in wild-type and PAWS1$^{-/-}$ cells over the course of 16 h of Wnt3A stimulation. Like Wnt3A-induced AXIN2 mRNA expression earlier (Fig 4B), we observed that a time course of Wnt3A stimulation induced an increase in AXIN2 protein levels in wild-type U2OS cells, while this increase was less pronounced in PAWS1$^{-/-}$ cells (Fig 9A). Since the Wnt3A-induced phospho- and total β-catenin levels in U2OS cell extracts did not appear to change significantly to account for the inhibition of Wnt signalling caused by loss of PAWS1, we analysed the cytoplasmic and nuclear distribution of β-catenin in wild-type, PAWS1$^{-/-}$, and PAWS1$^{WT}$ and PAWS1$^{F296A}$ rescue U2OS cells upon Wnt stimulation (Fig 9B). We monitored pSer$^{45}$, pThr$^{41}$/Ser$^{37}$/Ser$^{33}$, non-phospho Ser$^{45}$ and Thr$^{41}$ (so-called α-active β-catenin monoclonal antibody [40]) and total β-catenin levels. Compared to wild-type and PAWS1$^{WT}$ rescue cells, both basal and Wnt-induced levels of β-catenin-pSer$^{45}$, β-catenin-pThr$^{41}$/Ser$^{37}$/Ser$^{33}$, active β-catenin and to some extent total β-catenin in nuclear, but not cytoplasmic, fractions were markedly lower in PAWS1$^{-/-}$ and PAWS1$^{F296A}$ rescue cells (Fig 9B). This was particularly evident when we analysed the levels of active β-catenin in each fraction relative to that present in the cytoplasmic fraction of WT U2OS cells (Fig 9B, lower panel), suggesting that PAWS1 potentially promotes nuclear accumulation of β-catenin. In WT and PAWS1$^{WT}$ and PAWS1$^{F296A}$ rescue U2OS cells, PAWS1 was observed in both cytoplasmic and nuclear fractions, with slight enrichment in cytoplasmic fractions. However, more CK1α was detected in the nuclear fractions than cytoplasmic, except that there was less overall CK1α detected in PAWS1$^{-/-}$ and PAWS1$^{F296A}$ rescue cells compared to WT and PAWS1$^{WT}$ rescue cells (Fig 9B). These observations suggest that other FAM83 members that also bind CK1α probably account for the

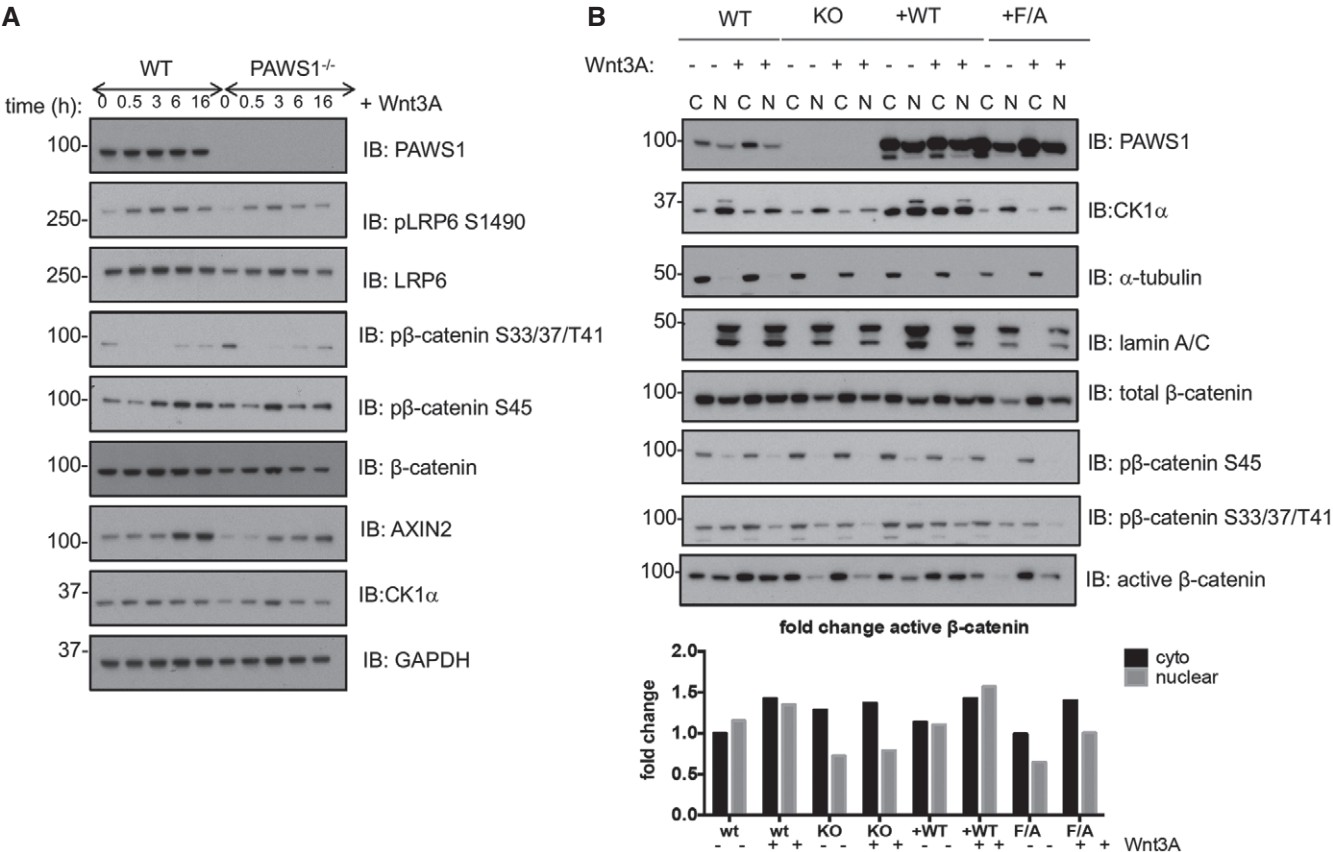

**Figure 9. PAWS1 promotes Wnt signalling through increased accumulation of nuclear active β-catenin.**

A  U2OS wild-type and PAWS1$^{-/-}$ cells were exposed to either Wnt3A or control medium for the indicated time points. Cell extracts were subjected to SDS–PAGE followed by Western blot analysis with the indicated antibodies.

B  U2OS wild-type (WT), PAWS1$^{-/-}$ (KO), PAWS1$^{WT}$ (+WT) and PAWS1$^{F296A}$ (F/A) rescue cells were exposed to either Wnt3A or control medium for 3 h followed by separation and preparation of cytoplasmic and nuclear fractions. The extracts were subjected to SDS–PAGE followed by Western blot analysis with the indicated antibodies. The lower panel represents the fold changes in active β-catenin intensities in each fraction relative to those seen in the cytoplasmic fraction of control WT U2OS cells. The intensities of the active β-catenin bands in each fraction were quantified by using the ImageJ software.

Source data are available online for this figure.

---

enhanced nuclear pool of CK1α [31]. Because most of the β-catenin protein is reported to be associated with the plasma membrane [41,42], we assessed the effect of PAWS1-loss on membrane-associated pool of β-catenin. We also found high levels of total β-catenin in the plasma membrane fractions and quite substantial amounts in the cytoskeletal fractions, when compared to cytoplasmic and nuclear fractions (Fig EV5E). However, neither the loss of PAWS1 nor Wnt3A stimulation changed the levels of β-catenin protein in both the membrane and cytoskeletal fractions (Fig EV5E). Under these conditions, very low levels of PAWS1 and CK1α were detected in the membrane and cytoskeletal fractions (Fig EV5E).

## Discussion

We have shown previously that PAWS1 modulates a subset of non-canonical SMAD4-independent genes in the BMP signalling pathway, yet has no effect on the canonical BMP pathway [1]. Consistent with this, we find here that PAWS1 does not affect canonical BMP signalling

in either *Xenopus* embryos or U2OS cells. In contrast, we demonstrate here that PAWS1 plays a significant role in regulating Wnt signalling in both developing *Xenopus* embryos and human cells. Furthermore, we find that the association between PAWS1 with CK1α is necessary to modulate Wnt signalling. The precise mechanisms for the coordinated actions of PAWS1 and CK1α in Wnt signalling, and perhaps beyond, remain to be established but our findings indicate that PAWS1 regulates cellular levels and subcellular localisation of CK1α protein, and potentially the phosphorylation of CK1α substrates. Collectively, our findings establish PAWS1 as a novel player in the Wnt signalling pathway and a critical regulator of CK1α.

It has long been known that CK1 isoforms play crucial roles in the Wnt/β-catenin signalling [28–30,34,35,43–46]. In fact, early observations that overexpression of CK1 isoforms in *Xenopus* embryos caused axis duplication suggested a positive role for CK1 in Wnt signalling [47]. However, it has since become evident that CK1 isoforms play both negative and positive roles in the Wnt/β-catenin signalling [29], and their activities may be regulated in a coordinated fashion [48]. The precise mechanisms by which CK1 isoforms are

regulated to allow them to execute these opposing outcomes in Wnt/β-catenin signalling remain elusive. From our data, it is evident that PAWS1 and CK1α coordinate to modulate Wnt signalling but the influence of CK1δ and CK1ε on Wnt signalling is not affected by PAWS1. This suggests that different CK1 isoforms target specific sets of substrates in cells and are uniquely regulated in driving specific cellular processes, including Wnt signalling. In this context, our findings that different FAM83 members, of which PAWS1 is a member, associate and co-localise with different CK1 isoforms suggest that perhaps coordinated actions of one or more FAM83 members may define CK1 isoform-specific substrates and actions in the Wnt/β-catenin signalling and beyond [31].

In the absence of Wnt signalling, it has been reported that autophosphorylation of the C-terminal domains of CK1δ and ε inhibits their catalytic activity, which is relieved upon Wnt stimulation by yet-unidentified phosphatases [49,50]. On the other hand, CK1α is constitutively active and therefore must be regulated in a different manner. One attractive model is the "scaffold or proximity effect" (reviewed in [29,51]). In this model, rather than directly regulating catalytic activity, scaffold proteins facilitate kinase activity by regulating substrate proximity to the kinase [52]. For example, Axin, which interacts with GSK3β, enhances β-catenin phosphorylation by CK1α and GSK3β, without altering CK1α kinase activity [53–56]. Similarly, PAWS1 may act as a scaffold by regulating the access of CK1α to its substrates. Consistent with this notion, we find that PAWS1 IPs from cells display CK1α catalytic activity and recombinant PAWS1 has no effect on intrinsic CK1α kinase activity *in vitro*. Our observations that while PAWS1 and Axin form robust complexes with CK1α and GSK3β, respectively, neither appears to bind β-catenin or other known Wnt pathway components robustly are consistent with the potential scaffolding roles of PAWS1 and Axin in facilitating the phosphorylation of CK1α and GSK3β substrates respectively, as protein kinases often engage with their substrates only transiently. It is still unclear precisely how PAWS1 acts in concert with CK1α and it remains feasible that PAWS1 regulates CK1α substrate availability in either a positive or negative fashion. From our data, the PAWS1:CK1α complex does not appear to regulate the phosphorylation of β-catenin at Ser[45]; however, the nuclear accumulation of β-catenin appears to require PAWS1:CK1α complex. We have also shown that all members of the FAM83 family can associate and co-localise with CK1α, while some also interact with CK1δ and CK1ε isoforms, suggesting that other FAM83:CK1 complexes may act redundantly or in a coordinated fashion to control Wnt/β-catenin signalling [31].

Future work will identify and characterise the PAWS1-dependent CK1α substrates that may be essential for mediating Wnt signalling, using phospho-proteomic approaches in wild-type and PAWS1$^{-/-}$ cells and address the molecular basis for interaction between PAWS1 and CK1α. Similarly, investigations are underway to understand how CK1α protein levels are regulated by PAWS1.

# Materials and Methods

### Plasmids, antibodies and reagents

All constructs were sequence-verified by the DNA Sequencing Service, University of Dundee (http://www.dnaseq.co.uk). They included PAWS1-GFP donor (DU48585), sense (DU48793) and anti-sense (DU48826) gRNAs, pCMV5-FLAG-PAWS1 (DU33274), FLAG-PAWS1 F296A/F300A (DU28026), FLAG-PAWS1 F296A (DU28024) and FLAG-PAWS1 F300A (DU28025), pBabe puro PAWS1 (DU33460), pBabe puro FAM83G F296A (DU28044), pBabe puro FAM83G F300A (DU28045), pBabe puro FAM83G F296A/F300A (DU28046), pcDNA5-Frt/TO-PAWS1-GFP (DU42816) and pcDNA5-Frt/TO-GFP (DU41455). For bacterial expression of proteins, PAWS1 wild type and mutants were cloned in pGEXP1 vectors. These include pGEXP-1 GST-PAWS1-6xHis (DU28293), pGEXP-1 GST-PAWS1 S614A (DU33321) and pGEXP-1 GST-PAWS1 F296A (DU28047). These constructs are available to request from the MRC-PPU reagents webpage (http://mrcppureagents.dundee.ac.uk) and the unique identifier (DU) numbers indicated above provide direct links to the cloning strategies and sequence details.

Antibodies included PAWS1 (DU33022), SMAD1 (DU19291), CK1α (Bethyl Laboratories #A301-991A-M), CK1ε (CST #12448), HA (Sigma HA-7 #H3663), FLAG-HRP (Sigma #A8592), GFP (Abcam, #ab6556), Myc tag (Sigma 9E10 #M5546), α-tubulin (ThermoScientific #MA1-80189, Sigma #T5168), AXIN2 (CST #76G6), AXIN1 (CST #2074), β-catenin (CST #D10A8; Santa Cruz Biotechnology #H-102), GSK3-α/β (Santa Cruz #sc-7291), GSK3β (CST #9315S), SLC5A10 (Abcam, #ab167156), active β-catenin (Millipore #05-665, CST D13A1 #8814), LRP6 (CST #C47E12), GAPDH (CST #D16H11), Lamin A/C (CST #2032) and HA-HRP (Roche 2013819001). All phospho-antibodies were purchased from Cell Signaling Technologies.

*Xenopus* Addgene constructs pCS2AxinMT (#16298) pCS2 GSK3β 9SA (#16333) were a kind gift from Peter Klein, Xi He kindly provided pCS2LRP6-m5 (#27283); Randall Moon pCS2 β-catenin-GFP (#16839), Super 8× TOPFlash (#12456), Super 8× FOPFlash (#12457), xCK1ε D128N (#16726); William Hahn and David Root CK1α (#23355); and Barry Gumbiner pCS2C-cadherin (#17023). The ORFs of *Xenopus* FAM83G (xPAWS1; Image Clone MGC:98851 Source Bioscience), CK1δ (image clone #5571715 Source Bioscience), CK1ε (Image Clone #6316339, Source Bioscience) were cloned into pENTR-D-TOPO (Invitrogen) vector and then transferred into pCS2_N-MT, pCS2_N-HA, pCS2_C-HA, pCS2_c-mCherryHA destination vectors using LR clonase (Invitrogen). The catalytically inactive forms of CK1α (CK1α KD, D136N) were generated by site-directed mutagenesis. pCS2nuc β-gal and pCS2 H2B_RFP were kind gifts from Anna Philpott (University of Cambridge) and Sean Megason (Harvard University), respectively.

Recombinant TGFβ$_1$ and BMP$_2$ were purchased from R&D Systems. Bortezomib was from LC Laboratories. Doxycycline was from Sigma and CHIR99021 from Axon. Recombinant kinases were generated by the MRC-PPU protein expression team. These include CK1α (DU329) and CK1α(D136N) (DU28371).

### *Xenopus* maintenance and manipulations

Regulations for the use of *Xenopus laevis,* as outlined in the Animals Scientific Procedures Act (ASPA) and implemented by the Home Office in the UK, were followed. *Xenopus laevis* embryos were obtained by *in vitro* fertilisation and staged according to Nieuwkoop and Faber (1975). Embryos were injected with capped RNA synthesised using SP6 mMessage mMachine kit (Invitrogen). Animal caps assays were as described [57]. Dissociated animal caps were plated onto μ-Slide 8-well glass bottom chamber slides (#80827, Ibidi) or

12-well μChamber slides (#81201) coated with 3 μg/ml recombinant human E-cadherin protein (#8505-EC-050; R&D Systems) in 0.7× Marc's Modified Ringers (MMR; 0.1 M NaCl, 2 mM KCl, 1 mM MgSO₄, 2 mM CaCl₂, 5 mM HEPES pH 7.8, 0.1 mM EDTA).

### In situ hybridisation

Whole-mount *in situ* hybridisation was performed as described [58], except the triethanolamine and acetic anhydride treatments were omitted. β-Gal staining was developed using Salmon Gal (Apollo Scientific #BIMB1026), as an alternative chromogenic substrate, which produces a light pink colour. Probes were made using DIG labelling mix (Roche, #11277037910), and developed using BCIP (Roche, #11383221001), and NBT (Roche, #11383213001). Images of stained embryos were captured using a Leica DFC310FX camera attached to a Leica M165FC microscope controlled by LAS V4.9 software.

### Immunofluorescence

Dissociated animal cap cells injected with β-catenin_GFP and/or xPAWS1_mCherry mRNAs were imaged live with a Zeiss LSM710 confocal microscope controlled by ZEN Black 2012 software. Post-acquisition analysis was performed using ZEN Black, Zen Blue 2012 and ImageJ software packages.

For the analysis of SMAD signalling, dissociated animal cap cells were fixed in MEMFA (0.1 M MOPS pH 7.4, 2 mM EGTA, 1 mM MgSO₄, 3.7% formaldehyde) for 10 min at room temperature and then washed 3 × 5 min in PBS. Cells were permeabilised with 0.5% Triton X-100 for 5 min followed by 3 × 5 min washes with PBS and then blocked in blocking buffer (PBS, 0.1% Tween-20, 1% BSA, 10% goat serum) for 1 h at room temperature. Primary (anti-P-Smad1, CST #9511) and secondary (anti-rabbit Alexa568, Invitrogen #A11031) were used at 1:1,000 in blocking buffer, each for 1 h, and then washed 3 × 5 min with PBS-T. Cells were mounted using Prolong Gold Antifade Mountant with DAPI (ThermoFisher, #P36935). Images were captured on Zeiss LSM710 confocal microscope as described above.

For PAWS1 and CK1α IF, cells were fixed with 4% paraformaldehyde in PBS for 10 min, and permeabilised with 0.5% Triton X-100 in PBS for 5 min. Coverslips were incubated in blocking buffer (3% BSA, 0.1% Triton X-100 in PBS) for 30 min, followed by anti-PAWS1 (1:250, Abcam, ab121750) and anti-CK1α (1:100, Santa Cruz, sc6477) antibodies in blocking buffer for 1 h. Cells were washed with 0.1% Triton X-100 in PBS and incubated with Alexa Fluor-conjugated secondary antibodies (ThermoFisher Scientific) and DAPI (1 μg/ml). Images were captured using a DeltaVision system (Applied Precision). Z-series were collected at 0.2 μm intervals and deconvolved using SoftWoRx (Applied Precision).

### Cell culture

U2OS osteosarcoma cells, HEK293 cells, mouse fibroblast L-cells that stably overexpress Wnt3A (ATCC, CRL-2647) or L cells (ATCC, CRL-2648) were grown in Dulbecco's Modified Eagle's Medium (DMEM; Gibco) supplemented with 10% (v/v) FBS (Hyclone), 2 mM ʟ-glutamine, 100 units/ml Penicillin and 100 μg/ml Strepto-mycin (Lonza). Flp-IN TRex U2OS cells were maintained in growth media supplemented with 15 μg/ml blasticidin and (100 μg/ml) zeocin. Cells were grown until confluent at 37°C in a humidified incubator at 5% CO₂. Prior to lysis, cells were stimulated as described in the figure legends. For transient transfections, cells were transfected with cDNA plasmids (2.5 μg per 10-cm dish unless indicated otherwise) using polyethylenimine (PEI) (Polysciences Inc.) in OptiMem (Gibco) medium as described previously [59].

### Preparation of cell extracts

Before being harvested, cells were rinsed in ice-cold PBS and extracted in lysis buffer containing 50 mM Tris–HCl pH 7.5, 1 mM EGTA pH 8, 1 mM EDTA pH 8, 1 mM activated Na₃VO₄, 10 mM Na β-glycerophosphate, 50 mM NaF, 5 mM Na Pyrophosphate, 270 mM Sucrose, 1% (v/v) Nonidet P-40 and a protease inhibitor cocktail (1 tablet per 25 ml buffer) (Roche). Cell lysates were clari-fied by centrifugation at 13,000 × *g* for 20 min at 4°C. Supernatant was recovered (cell extract), and protein concentration was deter-mined in a 96-well format using Bradford protein assay reagent (Pierce). For nuclear and cytoplasmic extraction, a NE-PER kit from Thermo Scientific (#78833) was used in accordance with the manu-facturer's instructions. For membrane and cytoskeletal protein extraction, a compartment protein extraction kit from Merck (#2145) was used according to the manufacturer's instructions.

*Xenopus* extracts were prepared by titrating either embryos or animal caps with 10 μl/embryo or 2 μl/animal cap of embryo lysis buffer (1% Triton X-100, 150 mM NaCl, 10 mM K-HEPES pH 7.5, 2 mM EDTA, Roche protease inhibitor cocktail). Lipids and yolk were removed by extracting the lysate with an equal volume of 1,1,2-Trichloro-1,2,2-trifluoroethane (FREON, Sigma Aldrich).

### Immunoprecipitation and immunoblotting

0.5–1 mg of total cell lysate protein was incubated with 10 μl beads (GFP-Trap® or anti-FLAG-M2 or 10 μg anti-CK1α antibody conju-gated to 10 μl Protein G-agarose beads (Abcam)) for 2 h at 4°C on a rotating platform. IPs were washed twice in lysis buffer supple-mented with 250 mM NaCl and once in Buffer A (50 mM Tris–HCl pH 7.5, 0.1 mM EGTA). Immunoprecipitated proteins were dena-tured in SDS sample buffer supplemented with 1% β-mercap-toethanol and then subjected to SDS–PAGE. After SDS–PAGE, the proteins were transferred to PVDF membrane. The membrane was blocked with 5% non-fat dry milk in TBS-T buffer (50 mM Tris–HCl, pH 7.5, 0.15 M NaCl and 0.1% Tween-20) for 30 min and then with primary antibody (1:1,000 in TBS-T with 5% BSA) overnight at 4°C. Blots were incubated for 1 h at room temperature with HRP-conjugated secondary antibodies (1:5,000 in 5% non-fat dry milk). Proteins were visualised using the ECL system (GE Healthcare).

*Xenopus* extracts were denatured in 4× loading buffer (LiCor, #928-40004) supplemented with 10% v/v β-mercaptoethanol, and either 1 embryo equivalent, or 10 animal cap equivalents were loaded per well of a 7.5% mini-Protean TGX gel (Bio-Rad #456-1025). Following electrophoresis, proteins were transferred using Trans-Blot Transfer System onto LF PVDF membranes. Blots were blocked with a 1:1 mixture of TBS and Odyssey Blocking Buffer (LiCor #92750000) for 1 h at room temperature and then incubated in primary antibodies diluted 1:1,000 in blocking buffer overnight at 4°C. Blots were washed 3 × 5 min in TBS-T (TBS, 0.1% Tween-20).

Secondary antibodies were added (goat anti-rabbit and anti-mouse Abs labelled with either IRDye 680LT or IRDye 800CW) diluted in TBS-T supplemented with 0.02% SDS for 1 h at room temperature. Blots were washed 3 × 5 min in TBS-T and then imaged on a LiCor Odyssey scanner using ImageStudio software (LiCor). The 700- and 800-nm channels were captured simultaneously at the setting of 5, the μ-setting was at 165 μm with the quality setting set at lowest.

## Generation of PAWS1$^{-/-}$ U2OS rescue cells stably expressing PAWS1$^{WT}$ or PAWS1 mutants

Retroviral pBabe-puromycin empty vector or vectors encoding PAWS1$^{WT}$ or indicated PAWS1 mutants (6 μg) were co-transfected with pCMV5-gag-pol (3.2 μg) and pCMV5-VSV-G (2.8 μg) into a 10-cm-diameter dish of ~70% confluent HEK293-FT cells. Briefly, cDNAs were added to 1 ml Opti-MEM medium to which 24 μl of 1 mg/ml PEI (diluted in 25 mM HEPES pH 7.5) was added. Following a brief vortexing (10 s) and incubation at room temperature for 20 min, the transfection mix was added dropwise to the HEK293-FT cells. 16 h post-transfection, fresh medium was added to the cells. 24 h later, the retroviral medium was collected and filtered through 0.45-μm filters. Target PAWS1$^{-/-}$ U2OS cells (~60% confluent) were infected with the optimised titre of the retroviral medium (1:100) diluted in fresh medium containing 8 μg/ml polybrene (Sigma) for 4 h. The retroviral infection medium was then replaced with fresh medium, and 20 h later, the medium was again replaced with fresh medium containing 2 μg/ml puromycin for selection of cells which had integrated the rescue constructs. The rescue cells were analysed by Western blotting as indicated.

## Generation of U2OS PAWS1-GFP knock-in cells (PAWS1$^{GFP/GFP}$) by CRISPR/Cas9

U2OS cells were transfected with vectors encoding a pair of guide RNAs (pBABED-Puro-sgRNA1 and pX335-CAS9-D10A-sgRNA2) targeting the last exon of PAWS1 (1 μg each) and a donor vector carrying the GFP construct fused to the last 129 amino acids of the last exon of PAWS1 (3 μg). 24 and 48 h post-transfection, cells were cultured in medium containing 2 μg/ml puromycin for selection of transfected cells. Cells were then cultured in normal medium, and transfection was repeated followed by 2 days of puromycin selection. GFP-positive cells were then sorted by flow cytometry, and single GFP-positive cells were plated on individual wells of two 96-well plates as described previously [60]. Viable clones were expanded, and integration of GFP at the target locus was verified by Western blotting and genomic sequencing.

## Generation of doxycycline-inducible Flp-IN TRex U2OS stable cells

Flp-IN TRex U2OS cells grown in 10-cm dishes were transfected with 1 μg pcDNA5-Frt/TO vector encoding PAWS1-GFP or GFP only and 9 μg pOG44 (Invitrogen), diluted in 1 ml OPTIMEM and 20 μl of 1 mg/ml PEI. 24 h post-transfection, cells were cultured in medium containing 15 μg/ml blasticidin and 50 μg/ml hygromycin B for 2–3 weeks (by replacing fresh medium every 2 days) until positive clones were selected, verified and expanded. Protein expression was induced with 20 ng/ml doxycycline for 16 h prior to lysis.

## Mass spectrometry

PAWS1$^{GFP/GFP}$ cells were lysed in DSP lysis buffer (40 mM HEPES pH 7.4, 120 mM NaCl, 1 mM EDTA pH 8, 10 mM Na-pyrophosphate, 50 mM NaF, 1.5 mM activated Na-orthovanadate, 1% (v/v) Nonidet P-40, protease inhibitor cocktail (1 tablet per 25 ml buffer) and 2.5 mg/ml DSP. The lysates were left on ice for 30 min, and cross-linking reactions were quenched by adding 62.5 mM Tris–HCl pH 7.5. Clarified lysates were filtered through a 0.45-μm filter and then subjected to immunoprecipitation using 10 μl of GFP-Trap® beads at 4°C overnight. IPs were washed twice in Lysis buffer supplemented with 500 mM NaCl and once in Buffer A (50 mM Tris–HCl pH 7.5, 0.1 mM EGTA). IPs were denatured in LDS sample buffer (Invitrogen) supplemented with 0.1 M DTT, incubated for 1 h at 37°C and then heated for 5 min at 95°C. Samples were filtered through a Spin-X column by centrifugation, resolved by 4–12% gradient SDS–PAGE, and stained with colloidal Coomassie blue overnight. Gels were destained in distilled H$_2$O until background staining was minimal. Sections of the gel were excised, trypsin digested, and peptides prepared for HPLC gradient fractionation and elution into a Thermo ScientificVelos Orbitrap mass spectrometer as described previously [61,62]. Ion assignments were conducted by *in silico* Mascot scoring (www.matrixscience.com), and peptide protein assignments were reported in Scaffold 4.3 (www.proteomesoftware.com).

## Dual luciferase reporter assays

Cells (2 × 10$^5$) were plated in 6-well plates. 24 h later, 500 ng of Super TOPFlash or Super FOPFlash and 10 ng of Renilla-luciferase [63] plasmids were co-transfected using PEI and OPTI-MEM. For transient PAWS1 expression, 500 ng of pBabe PAWS1 (DU33460) was included to the transfection mix. 24 h later, cells were treated with L-conditioned medium or L-Wnt3A-conditioned medium or 20 mM LiCl for 16 h, washed twice with PBS and lysed in passive Lysis buffer (Promega, #E194A). Firefly and Renilla luciferase activities were measured as described [60,64]. The firefly luciferase counts were normalised to Renilla luciferase counts, which represented a measure of transfection efficiency in each sample.

## Quantitative PCR and primers

Total RNA was isolated from cells using the QIAGEN RNeasy kit. Real-time quantitative reverse transcription PCR (qRT–PCR) was performed using 1 μg of isolated RNA and the SuperScript cDNA kit (Bio-Rad) according to the manufacturer's protocol. qRT–PCRs were performed in triplicate in a 12 μl reaction volume. Each reaction included 0.5 μM forward primer, 0.5 μM reverse primer, 50% SYBR green master mix (Bio-Rad), and cDNA equivalent to 1 ng/μl of RNA in a CFX96 real-time system qRT–PCR machine (Bio-Rad). The data were normalised to the geometrical mean of the β-actin gene and analysed as described previously [62]. Primers were designed using PerlPrimer and purchased from Invitrogen: AXIN2 forward: TACACTCCTTATTGGGCGATCA; reverse: TTGGCTACTCGTAAAGTTTTGGT. CYCLIN D1 forward: GCTGCGAAGTGGAAACCATC; reverse: CCTCCTTCTGCACACATTTGAA. CK1α forward: GAGGCAGCTATTCCGCATTC; reverse: TGGGGAGAAACAAATGCTGC. Actin forward: CCAACCGCGAGAAATGACC; reverse: GGAGTCCATC

ACGATGCCAG. For RT–PCR in *Xenopus* laevis, total RNA was extracted from either embryos or animal caps using TRIzol reagent (Ambion) according to the manufacturer's instructions. 1 µg of RNA was used in the cDNA synthesis reaction using random hexamers and MMLV-RT (Promega). Quantitative PCR was performed using LightCycler 480 SYBR Green Master I (Roche) using standard run template conditions on a LightCycler 480 machine (Roche). Primers used are as follows: Histone H4 forward: CGGGATAACATTCAGGG-TATCACT; Histone H4 reverse: ATCCATGGCGGTAACTGTCTTCCT; Siamois forward: AACCACTTTCCACTCTCCCC; Siamois reverse: AGAAGTCAGTTTGGGTAGGGC; Msx1 forward: GCAGGAACATCAC ACAGTCC; Msx1 reverse: GGGTGGGCTCATCCTTCT; Vent1 forward: GCATCTCCTTGGCATATTTGG; Vent1 reverse: TTCCCTT CAGCATGGTTCACC; Vent2 forward: TGAGACTTGGGCACTGTCTG; Vent2 reverse: CCTCTGTTGAATGGCTTGCT; Chordin forward: AACTGCCAGGACTGGATGGT; Chordin reverse: GGCAGGATTTAGA GTTGCTTC. Data were normalised to Histone H4.

### *In vitro* kinase assays and phospho-mapping

25 µl reactions were set up using 200 ng of kinase, 2 µg of substrate in buffer containing 50 mM Tris pH 7.5, 0.1 mM EGTA, 10 mM MgAc$_2$, 2 mM DTT and 0.1 mM [$^{\gamma 32}$P]-ATP (~500 cpm/pmol). Reactions were performed at 30°C for 30 min and stopped by adding 10 µl LDS buffer with 5% β-mercaptoethanol and heating at 95°C for 5 min. Samples were resolved by SDS–PAGE, and gels were stained with Instant blue (Expedeon) and dried. Radioactivity was analysed by autoradiography. Kinase assays against substrate peptide (CK1tide, DSTT EP5630) were performed as described previously [33].

For phospho-mapping, [$^{\gamma 32}$P]-phosphorylated PAWS1 was subjected to trypsin digestion and peptides were reconstituted to a final concentration of 5% ACN/0.1% TFA. The peptides were injected into a C$_{18}$ column equilibrated with 0.1% TFA with a linear ACN/0.1% gradient at a flow rate of 0.2 ml/min, and 100 µl fractions were collected. The major eluted peptides were analysed on OrbiTrap Velos LCMS system. To determine the phosphorylated residue in each [$^{\gamma 32}$P]-labelled peptide, the peptides were immobilised on a Sequelon-AA membrane and subjected to solid-phase Edman degradation as described previously [65].

### RNA interference (RNAi)

U2OS cells were seeded at ~1 × 10$^5$ cells per well in 6-well cell culture plates. Cells were transfected with 5, 10 or 20 nM of the relevant small interfering RNA (siRNA) oligonucleotides using 4 µl Lipofectamine RNAiMAX (Invitrogen) or TransFectin (Bio-Rad) per well in Opti-MEM reduced serum medium (Gibco). After 16 h, the transfection media were removed and replaced with normal DMEM culture medium. Cells were lysed 48 h post-transfection. All siRNA oligonucleotides were purchased from Dharmacon (Product refs: J-003957-13, J-003957-14, J-003957-15, J003957-16).

### Statistical analysis

Data are presented as mean ± standard error of the mean. Statistical significances of differences between experimental groups were assessed using two-way ANOVA with *post hoc* Bonferroni

correction, unless stated differently in figure legends, using Prism6 software. Differences in means were considered significant if *P* < 0.05.

**Expanded View** for this article is available online.

### Acknowledgements
We thank L. Fin, J. Stark, and A. Muir for help with tissue culture, the staff at the Sequencing Service (School of Life Sciences, University of Dundee, UK) for DNA sequencing, and the protein & antibody production and cloning teams at the Division of Signal Transduction Therapy (DSTT; University of Dundee) coordinated by H. McLauchlan and J. Hastie. KW and TDC are supported by the UK MRC Career Development Fellowship. PB and LDH are supported by the U.K. MRC Prize PhD studentship. GPS is supported by the U.K. Medical Research Council (MC_UU_12016/3) and the pharmaceutical companies supporting the DSTT (Boehringer-Ingelheim, GlaxoSmithKline and Merck-Serono). JCS, KSD and FC are supported by the Francis Crick Institute, which receives its core funding from Cancer Research UK (FC001157), the UK Medical Research Council (FC001157) and the Wellcome Trust (FC001157).

### Author contributions
PB performed most of the experiments in mammalian cells, collected and analysed data and contributed to the editing of the manuscript. KSD conceived and performed most of the experiments in *Xenopus* embryos and animal caps, wrote parts of the manuscript and collected and analysed data. KZLW, FC, TDC, LDH and JVo performed some of the experiments. TJM designed strategies and developed methodologies for and generated CRIPSR/Cas9-knock-out and knock-in constructs. NTW performed cloning and mutagenesis for most of the constructs. JVa, RG and DGC collected and analysed mass spectrometry data. JCS and GPS conceived the project and wrote the manuscript.

### Conflict of interest
The authors declare that they have no conflict of interest.

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
