## [Review Process File · EMBO Reports]

PAWS1 controls Wnt signalling through association with Casein Kinase 1a

Polyxeni Bozatzki, Kevin S Dingwell, Kevin ZL Wu, Fay Cooper, Timothy D Cummins, Luke D Hutchinson, Janis Vogt, Nicola T Wood, Thomas J Macartney, Joby Varghese, Robert Gourlay, David G Campbell, James C Smith, Gopal P Sapkota

Review timeline:

Submission date:	13 July 2017
Editorial Decision:	11 August 2017
Revision received:	5 November 2017
Editorial Decision:	8 December 2017
Revision received:	23 January 2018
Editorial Decision:	31 January 2018
Revision received:	5 February 2018
Accepted:	7 February 2018

Editor: Esther Schnapp/Martina Rembold

Transaction Report:

1st Editorial Decision

11 August 2017

Thank you for the submission of your research manuscript to our journal. We have now received the full set of referee reports that is copied below. Since Esther Schnapp is currently traveling I have taken over the handling of your manuscript for the moment.

As you will see, the referees acknowledge the potential interest of the findings. However, all referees also point out several technical concerns and have a number of suggestions for how the study should be strengthened. In particular, all referees ask for further experiments to address if PAWS1 sequesters CK1 α away from the destruction complex. I think that all suggestions should be addressed and all control experiments have to be provided with the exception of point 3 from referee #3. Upon further discussion with the referees we concluded that it is certainly of interest to analyze the effect of PAWS1 on other CK1-dependent processes but these experiments might be too far reaching and are not required.

Given these constructive comments, we would like to invite you to revise your manuscript with the understanding that the referee concerns (as detailed above and in their reports) must be fully addressed and their suggestions taken on board. Please address all referee concerns in a complete point-by-point response. Acceptance of the manuscript will depend on a positive outcome of a second round of review. It is EMBO reports policy to allow a single round of revision only and acceptance or rejection of the manuscript will therefore depend on the completeness of your responses included in the next, final version of the manuscript.

Revised manuscripts should be submitted within three months of a request for revision; they will otherwise be treated as new submissions. Please contact us if a 3-months time frame is not sufficient

for the revisions so that we can discuss the revisions further.

Supplementary/additional data: The Expanded View format, which will be displayed in the main HTML of the paper in a collapsible format, has replaced the Supplementary information. You can submit up to 5 images as Expanded View. Please follow the nomenclature Figure EV1, Figure EV2 etc. The figure legend for these should be included in the main manuscript document file in a section called Expanded View Figure Legends after the main Figure Legends section. Additional Supplementary material should be supplied as a single pdf labeled Appendix. The Appendix includes a table of content on the first page, all figures and their legends. Please follow the nomenclature Appendix Figure Sx throughout the text and also label the figures according to this nomenclature. For more details please refer to our guide to authors.

We now strongly encourage the publication of original source data with the aim of making primary data more accessible and transparent to the reader. The source data will be published in a separate source data file online along with the accepted manuscript and will be linked to the relevant figure. If you would like to use this opportunity, please submit the source data (for example scans of entire gels or blots, data points of graphs in an excel sheet, additional images, etc.) of your key experiments together with the revised manuscript. Please include size markers for scans of entire gels, label the scans with figure and panel number, and send one PDF file per figure or per figure panel.

As part of the EMBO publication's Transparent Editorial Process, EMBO reports publishes online a Review Process File to accompany accepted manuscripts. This File will be published in conjunction with your paper and will include the referee reports, your point-by-point response and all pertinent correspondence relating to the manuscript.

I look forward to seeing a revised version of your manuscript when it is ready. Please let me know if you have questions or comments regarding the revision.

REFeree REPORTS

Referee #1:

Comments on manuscript number: EMBOR-2017-44807-T: "PAWS1 controls Wnt signalling through association with Casein Kinase 1 α " by Bozatzi/Dingwell and colleagues.

The manuscript by P Bozatzi, KS Dingwell and colleagues reports as major finding that the hitherto poorly characterized FAM83 family member PAWS1 interacts with casein kinase 1 α (CK1 α), and as a consequence, positively regulates Wnt/ β -catenin pathway activity. By a series of well performed and in most cases thoroughly controlled experiments which include complementary gain-of-function and loss-of-function experiments in *Xenopus* embryos and in human cell lines, the authors convincingly establish the role of PAWS1 as a new Wnt/ β -catenin pathway component. The authors determine where in the Wnt pathway PAWS1 acts, and identify by a proteomic approach CK1 α as a PAWS1 interaction partner. PAWS1 and CK1 α appear to regulate each other's abundance and/or activity, and intracellular location. Through its influence on CK1 α , PAWS1 apparently affects phosphorylation, nuclear translocation and transcriptional activity of β -catenin. Concerning the mode of action of PAWS1, the only major shortcoming of the study is that the relationship between PAWS1 and components of the destruction complex other than CK1 α was not addressed. Nonetheless, the authors already provide interesting and important new insights into the

intracellular processing of Wnt signals. Wnt growth factor signalling is highly relevant throughout embryonic development and in adult tissue homeostasis of metazoans. It is also strongly implicated in the genesis of a variety of human cancers. Yet, mechanistic details of Wnt signal transduction and key players are still unknown. Therefore, the new findings reported here will be of strong appeal to scientists with an interest in signalling mechanisms, developmental biology, and tumour biology.

Specific comments:

1. Figure 3F-H: Protein levels of PAWS1 mutants should be monitored by Western blotting to exclude the possibility that differences in Siamois induction are due to unequal expression.
2. Figure 4D,E and Figure 5: Through their epistasis experiments the authors very convincingly establish that PAWS1 acts at the level of the destruction complex and β -catenin. The study would become much more informative and could easily provide considerably deeper mechanistic insights if the authors checked whether the PAWS1 complex in addition to CK1 α contains components of the destruction complex (AXIN1, GSK3, APC) and β -catenin, and how this might change upon Wnt stimulation. The authors have all the tools for this at hand and the mass spectrometry data may already provide some clues.
3. Figure 4E,F: The role of GSK3 in PAWS1-induced gene regulation is confusing. The reduced potential to overcome PAWS1-induced upregulation of Siamois could be explained by insufficient GSK3 expression. However, in view of their model that PAWS1 modulates β -catenin levels and activity through CK1 α and the results shown in Figure 3I, how do the authors explain that PAWS1-deficiency abrogates the ability of the GSK3 inhibitor to induce TOPFLASH activity?
4. Figure 6D: Information about PAWS1 and CK1 α mRNA levels in the panel of cancer cell lines need to be provided to corroborate positively correlated expression of PAWS1 and CK1 α specifically at the protein level.
5. Figure 7C-F. The data cannot be unambiguously interpreted and the regulatory relationship between PAWS1 and CK1 α remains unclear. The experimental set up in Figures 7C/F suggests that CK1 α is a negative regulator of PAWS1 independent of its kinase activity because CK1 α can neutralise the axis-inducing capacity of PAWS1. This contrasts with the remainder of the manuscript where the authors rather seem to argue that PAWS1 inhibits CK1 α to facilitate activation of Wnt signalling. To make things consistent and more informative it should be investigated how PAWS1 co-expression affects the ability of CK1 α to induce a secondary axis when CK1 α is injected at higher amounts. Likewise, it needs to be shown whether or not kinase dead CK1 α has axis inducing abilities.
6. Figure 8A: I disagree with the authors' description of the data shown. To me, there appear to be differences in total amounts of β -catenin and the dynamics of phosphoforms of β -catenin and LRP6 in PAWS1 wild-type and PAWS1-deficient cells. I recommend to quantify and normalize the Western blot data to facilitate the comparison of total β -catenin, LRP6, and their phosphorylated derivatives under the different experimental conditions.
7. Figure 8B: In contrast to the immunofluorescence analyses shown in Figure 5F it appears that PAWS1 and CK1 α do not completely colocalize and are inversely distributed in cytosolic and nuclear fractions of wild-type cells. This differential distribution seems to be distorted upon re-expression of PAWS1 at higher than physiological levels. Furthermore, Wnt3a stimulation seems to increase cytoplasmic PAWS1 while decreasing nuclear CK1 α in wild-type cells. Again, quantification of the Western blot results should be helpful for the interpretation of the data. Possibly, results obtained under conditions of PAWS1 overexpression need to be re-evaluated concerning the potential mechanisms of CK1 α regulation.
8. The authors propose that the precise balance between PAWS1 and CK1 α is of critical importance for the mutual control of their stability and their modulation of Wnt signalling. In this regard the observation that PAWS1 is not the only FAM83 family member which interacts with CK1 α , could be highly relevant. This may result in a competitive setting where the consequences of PAWS1 overexpression and deficiency are nearly impossible to control and to unequivocally interpret. Functional redundancy could also be an issue. The authors should at least mention the results of the

accompanying manuscript by Fulcher et al. in sufficient detail (all FAM83 family members interact and colocalize with CK1 α) and discuss the potentially confounding impact on the study presented here.

Minor issues:

9. Abstract: line 8 word duplication „that that"
10. Figure 2C: What is the prominent signal that appears in the P-SMAD3 Western blot upon BMP treatment of cells?
11. Figure 3E: The analysis of Chordin expression should be mentioned in the manuscript text, not only in the figure.
12. Figure 8: The spelling of wnt3A/WNT3A is inconsistent in panels A and B.
13. Supplemental Figure S1: There is a discrepancy between the main manuscript text and the figure legend concerning the amount of RNA injected (1 ng or 500 pg?).
14. Supplemental Figure S2: There is a mislabelling of the panels in the figure legend.
15. Supplemental Figure S2: Please adjust the column width in Figure S2 panel D.
16. Supplemental Figure S4: Please align labels of time points and lanes in panel B.

Referee #2:

This manuscript introduces a FAM83 family member, PAWS1, as a regulator of the Wnt pathway. The author show by both overexpression and depletion that PAWS1 has a strong influence on Wnt signalling, monitored through expression of reporter gene, of known direct target gene, as well as the ability to induce a secondary axis in *Xenopus* embryos. The phenotypes are impressive, posing PAWS1 as an important regulator of the pathway.

The author further show that PAWS1 binds casein kinase 1alpha, and that this interaction accounts for the role of PAWS1 in Wnt signalling. These experiments, which include the use of point mutated PAWS1 variants, are extremely convincing.

The function of PAWS1 is narrowed down by showing that a) CK1a phosphorylates PAWS1, yet the phosphorylated residues are not required for PAWS1 activity; b) PAWS1 has no effect on CK1a phosphorylation activity toward a generic substrate, and c) the presence of PAWS1 has a striking stabilization effect on CK1a levels.

Altogether, this is an impressive piece of work, combining functional experiments in *Xenopus* study with beautiful biochemistry in cell lines, using perfect controls (PAWS1 *-/-* cells, knockin rescues,...). The data are clean and fully convincing.

What remains a weak aspect of the manuscript is the actual mode of action of PAWS1:

- 1) What is the impact of PAWS1-CK1a interaction? I am certainly aware that a complete biochemical explanation would be beyond the scope of this first study, but a simple set of IP experiments would clarify this issue right away by discriminating between two obvious alternatives: Is PAWS1 sequestering CK1a away from the Axin complex, or is the PAWS1 recruited to the Axin complex? In the first scenario, the Axin-CK1a interaction should be decreased in the presence of PAWS1. In the second scenario, both CK1a and PAWS1 should coprecipitate with Axin.
- 2) No explanation for destabilization of PAWS1 and CK1a in the absence of their partner is presented. Supplemental Fig S4 panel A, in wt cells there seem to be a stabilization of both PAWS1 and CK1a upon inhibition of the proteasome. Wouldn't this suggests that ubiquitination/ proteasomal degradation is involved?
- 3) The authors propose that the right levels of PAWS1 and CK1a are required for proper regulation of Wnt signalling. This hypothesis is based in particular on the analysis of induction of secondary axis in *Xenopus*. Page 9: "human CK1a can also induce a secondary axis in a dose dependent manner in *Xenopus* embryos (Fig 7C). However, when co-expressed with PAWS1, the axis induction was blocked (Fig 7D)." Yet, this statement is not supported by the data, because the dose of CK1a mRNA used in 7D (50ng) is not sufficient alone to induce secondary axis. The experiment

should be performed with the high CK1 α dose (200ng). Panel F: Doses for PAWS1 and CK1 α kinase dead mRNA not indicated.

Further issues:

4) Related to point 1. PAWS1 has no effect on CK1 α activity on a peptide substrate, but could have an effect on endogenous substrates e.g. within the Axin complex. Please comment.

5) Fig8B and bottom of page 10: A kit is used to separate cytoplasmic and nuclear fractions, from which conclusions are drawn about the effect of PAWS1 on cytoplasmic/nuclear b-catenin. However, most b-catenin is supposed to be present at the plasma membrane. In which fraction are membranes recovered? Please include a plasma membrane marker (cadherin, LRP6 or other) and adjust interpretation according to result.

6) Role of PAWS1 is BMP signalling: The data presented here argue that PAWS1 does not influence this pathway, yet such role was previously proposed by the authors. I would minimally expect a short explanation in the discussion.

Minor comments

Fig3A. The specificity of the "active b-cat" antibody is not well established. In fact, in figure 8B, patterns for "active" and total b-cat may be identical, except for the difference in strength of the signal. I suggest to interpret these data with caution.

Fig5F. PAWS1-CK1 colocalization is not quite obvious: Including a high magnification and pointing to colocalizing/non-colocalizing spots would help. One would also like to see a comparison of the degree of colocalization with another unrelated cytoplasmic protein, e.g. PAWS1/GAPDH.

Fig6C: In CK1 α KD, PAWS1 is clearly decreased, but still half is left, which implies that PAWS1 can subsist in the absence of CK1 α . Please comment.

Fig S2E: How come CK1 α is not visible in input?

Legend Fig S2: Correct reference to panels (a,b,b,c,d)

FigS3E: PAWS1 phosphorylation: not required for ectopic axis duplication. But could be important for endogenous regulation?

FigS4B: what is LC3?

Referee #3:

PAWS CK1 review

The manuscript from Bozatz et al describes a role for PAWS1/FAM83G in regulation of Wnt signaling. In *Xenopus* embryos and U2OS cells, over-expression of PAWS1 activates β -catenin signaling, while knockout inhibits signaling. In search of mechanism, they find strong interaction with CK1 α , a known regulator of Wnt signaling. Epistasis studies place PAWS1 at or near the destruction complex. Point mutations in PAWS1 that block interaction with CK1 α also block the Wnt/ β -catenin stimulating activity of PAWS1. The data taken together suggest that PAWS1 acts like a CK1 α sink that can sequester active CK1 away from relevant targets in the Wnt pathway. However, here there is no clear evidence for changes in the phosphorylation of the myriad CK1 targets. Whether PAWS affects other CK1 dependent processes is not well addressed. The quality of the data on whole is good, with specific exceptions noted below. The mechanism of PAWS1 activation remains unsettled. The paper does not settle on mechanism, which will reduce its impact in the field. It is also odd, and not particularly well addressed, that this must not be a core mechanism in Wnt signaling since other Wnt-responsive cell lines do not express PAWS.

A few key questions remain.

Key experiments I would like to see:

1. They show that active or inactive CK1 α reverses the signaling effect of PAWS1. Does CK1 δ or CK1 ϵ expression also reverse this signaling effect?
2. Does PAWS1 expression change the amount of CK1 isoforms co-immunoprecipitating with axin or Disheveled? That might be a simple experiment to see if PAWS1 simply removes CK1 α from the destruction complex leading to stabilization of β -catenin.
3. Does PAWS expression alter other CK1 dependent processes? This is not yet well addressed.

General comment: plunger plots are thankfully going out of fashion and should not be used here. Experiments with small samples sizes should use scatter-plots, dot plots or similar methods that permit direct evaluation of the distribution of the data. C.f. Weissgerber et al. (2015) Beyond Bar and Line Graphs: Time for a New Data Presentation Paradigm. PLoS Biol 13(4): e1002128. doi:10.1371/journal.pbio.1002128

Other questions and issues:

As a comment, CK1 plays multiple roles in Wnt signaling including phosphorylation of β -catenin, Disheveled, LRP6, LEF1 and APC. These papers should be referenced and considered in the discussion.

Fig 1C how much hPAWS mRNA was injected?

Fig 1E&F n=3 seems like a very small n to have much confidence in the results.

Fig 3A needs molecular weight markers. I'm confused by what Fig 3B and C are quantitating. It would help to specify which antibody is used in each case. The quantitation in fig 3C doesn't seem to match with what I can see in the "active β -catenin" blot of Fig 3A.

It is surprising in Fig 3A: why does xPAWS stabilize β -catenin_GFP but has no noticeable effect on endogenous "active" β -catenin? Would it be helpful to also probe for total β -catenin?

Fig 3D: the figure legend states "Wnt activity, stimulated with 50 μ M of the GSK3-b inhibitor CHIR99021, induces stabilization and nuclear localization of β -catenin_GFP in the absence of xPAWS1 expression." However, that is not visible in the figure, second row. Unless the authors are referring to some faint nuclear signal very unlike that seen with xPAWS1 co-expression. This requires clarification.

The figure legend is unclear - is figure 3E done with tagged or untagged xPAWS? How much mRNA of each construct was injected?

Fig 3F-H: is it possible that failure to induce axis and Siamois expression is due to production of an unstable protein? Without evidence of protein expression one cannot conclude much from negative results in DUF, 151, etc... mutants.

Odd that PAWS doesn't do nearly so much in HEK293 cells as it does in U2OS. This is consistent with the data in fig 6D, where PAWS1 is low or undetectable in a number of cell lines that presumably are responsive, like HEK293 cells, to Wnt ligands. It is known that other CK1 genes (CK1 δ , CK1 ϵ) can also prime β -catenin for destruction. Have the authors probed the same blots for CK1 δ and CK1 ϵ ?

In these types of signaling assays, the amount of plasmid transfected is important to note. Please include this information, preferably in the figure legends.

In Fig 4, the loss of signaling in U2OS cells knocked out for PAWS is impressive. It would be important to test if this effect was specific to the Wnt pathway, e.g. by testing other signaling pathways to test if they were similarly affected. Clearly, some cells without PAWS (HEK293 cells) are highly responsive to Wnt signaling. The NF-AT experiments in S5 are not compelling (see note below).

In figure 5D, it appears there is near complete depletion of CK1 α in the flowthrough after expression of WT PAWS1. That is impressive. It would be interesting (not essential) to know the fold overexpression of PAWS1 above endogenous.

Fig 7A and 7B are strong evidence for the importance of the PAWS1-CK1 α interaction. Fig 7B: please clarify in text or figure legend, these are transient transfections?

I find fig 8 both difficult to interpret, and not illuminating. First, it's unclear if this was a one-off, or a multiply replicated experiment with statistical confidence in the small changes. Second, while it seems to confirm their prior data on β -catenin regulation, it doesn't help me understand mechanism. I don't think this adds tremendously to my understanding.

Minor: please include molecular weight indicators on all SDS-PAGE gels and immunoblots.

Supplemental figure 2D and E legends need to be corrected.

Supplemental figure 2E: Why is no CK1 α present in the input? It's visible in U2OS cells in many other figures. It's important because the absence of CK1 α in the flow through might have suggested complete depletion of CK1 α , but not if we can't see it in the input.

The Mass spectrometry top hits should all be listed in a table in supplemental data so that we can assess if there were other high probability interactors.

Supplemental Fig 5 lack a control for CK1 activity. If the model is, increased CK1 activity drives NFAT to the nucleus, they need to show this happens in their system, either by expression of CK1, or inhibition of CK1 in the presence of the ionophore. Otherwise we can't interpret this negative result. I also wonder if the dose of ionophore is too high, since there looks to be a lot of apoptosis in the samples?

1st Revision - authors' response

5 November 2017

Responses to Reviewers' comments: The reviewer's comments are *italicized* and our responses appear as non-italicized fonts. New data and figures are indicated with **bold face fonts**.

Referee #1:

Comments on manuscript number: EMBOR-2017-44807-T: "PAWS1 controls Wnt signalling through association with Casein Kinase 1 α " by Bozatzki/Dingwell and colleagues.

*The manuscript by P Bozatzki, KS Dingwell and colleagues reports as major finding that the hitherto poorly characterized FAM83 family member PAWS1 interacts with casein kinase 1 α (CK1 α), and as a consequence, positively regulates Wnt/ β -catenin pathway activity. By a series of well performed and in most cases thoroughly controlled experiments which include complementary gain-of-function and loss-of-function experiments in *Xenopus* embryos and in human cell lines, the authors convincingly establish the role of PAWS1 as a new Wnt/ β -catenin pathway component. The authors determine where in the Wnt pathway PAWS1 acts, and identify by a proteomic approach CK1 α as a PAWS1 interaction partner. PAWS1 and CK1 α appear to regulate each other's abundance and/or activity, and intracellular location. Through its influence on CK1 α , PAWS1 apparently affects phosphorylation, nuclear translocation and transcriptional activity of β -catenin. Concerning the mode of action of PAWS1, the only major shortcoming of the study is that the relationship between PAWS1 and components of the destruction complex other than CK1 α was not addressed. Nonetheless, the authors already provide interesting and important new insights into the intracellular processing of Wnt signals. Wnt growth factor signalling is highly relevant throughout embryonic development and in adult tissue homeostasis of metazoans. It is also strongly implicated in the genesis of a variety of human cancers. Yet, mechanistic details of Wnt signal transduction and key players are still unknown. Therefore, the new findings reported here will be of strong appeal to scientists with an interest in signalling mechanisms, developmental biology, and tumour biology.*

Specific comments:

1. *Figure 3F-H: Protein levels of PAWS1 mutants should be monitored by Western blotting to exclude the possibility that differences in Siamese induction are due to unequal expression.*

Response: We have added a **panel (J) to Fig EV1** (All Supplementary Figures have been updated with Expanded View figures in line with Embo rep guidelines) with the western blot showing the expression of the Myc-tagged(MT)-xPAWS1 mutants.

2. *Figure 4D,E and Figure 5: Through their epistasis experiments the authors very convincingly establish that PAWS1 acts at the level of the destruction complex and β -catenin. The study would become much more informative and could easily provide considerably deeper mechanistic insights if the authors checked whether the PAWS1 complex in addition to CK1 α contains components of the destruction complex (AXIN1, GSK3, APC) and β -catenin, and how this might change upon Wnt stimulation. The authors have all the tools for this at hand and the mass spectrometry data may already provide some clues.*

Response: We have taken the reviewer's, and those of other reviewers', suggestions on board in order to gain deeper mechanistic insights and addressed the issues in a number of ways:

i. We undertook IP/Western strategy in unstimulated and Wnt3A (3h) stimulated U2OS cells to dissect any changes in the destruction complex components in the presence or absence of PAWS1 (**new Fig 8; new Fig EV5A**). No components of the destruction complex that we tested were detectable in endogenous CK1 α IPs from WT or PAWS1-KO cells (**new Fig 8; Fig EV5A**). As expected, PAWS1 was identified in CK1 α IPs from WT cells but not PAWS1-KO cells (**new Fig 8; Fig EV5A**), and the levels of PAWS1 in CK1 α IPs in WT cells did not change by Wnt3A treatment. Indeed, we are not aware of any previous reports that have experimentally demonstrated CK1 α as a constituent of the destruction complex. In Axin1 IPs, we were able to detect GSK3 α/β , and very low levels of β -catenin, although Wnt3A treatment or PAWS1-status did not appear to change the complexes (**new Fig 8**). Similarly, in β -catenin IPs, we were able to detect low levels of GSK-3 β , as well as low levels of CK1d and CK1e, suggesting that these kinases may be involved (rather than CK1 α) in mediating β -catenin phosphorylation at Ser⁴⁵. These experiments were not definitive in establishing whether PAWS1 actually impacts the destruction complex directly. Therefore, we decided to undertake an unbiased proteomic approach as described below.

ii. We undertook an unbiased proteomic approach to identify interactors of PAWS1-GFP expressed in U2OS cells under unstimulated and Wnt3A (3h) stimulated conditions (**new Fig EV5B-C**). However, other than CK1 α and CK1 α -like, no major components of either the Wnt pathway or the destruction complex, including Axin1, GSK-3, APC and Dishevelled, were identified in the GFP-PAWS1 IPs under both control and Wnt3A-stimulated conditions (**new Fig EV5C**). While some PAWS1 interactors, such as PLOD1 and many 14-3-3 proteins, were substantially enriched in Wnt3A stimulated conditions, albeit with much lower abundance compared to CK1 α and CK1 α -like, these still need to be validated further and their roles in Wnt signalling are not well established (**new Fig EV5C**). As 14-3-3 proteins are binders of phospho-proteins, we analysed all phospho-residues on PAWS1 by mass-spectrometry. Interestingly, some phospho-residues on PAWS1, namely Ser127, Ser634 and Ser726, appeared to be enriched by Wnt3A stimulation (**new Fig EV5D**). Clearly, these need to be validated at the endogenous level and the functional relevance of these is unclear.

iii. We can detect CK1 α kinase activity (as measured by an in vitro kinase assay against a peptide substrate) in PAWS1 IPs from WT U2OS cells but not PAWS1-KO cells, suggesting PAWS1 itself is unlikely to inhibit intrinsic CK1 α kinase activity (**new Fig 6A**). Therefore, the most plausible explanation for the role of PAWS1-CK1 α complex in Wnt signaling is that this complex determines the phosphorylation (or lack thereof) of crucial CK1 α substrate(s), most likely in the cytoplasm where the complex is mostly localized, that determine(s) the nuclear translocation of active β -catenin (downstream of the destruction complex). The precise mechanisms by which active β -catenin translocates to the nucleus and whether CK1 α plays a role remains poorly characterized. As protein kinase-substrate interactions are transient in nature, IP/MS and/or IP/Western approaches, like the ones we undertook above, are therefore unlikely to yield mechanistic insights or key spatio-temporally regulated substrates. Only a comprehensive phospho-proteomic analysis looking at PAWS1-dependent CK1 α substrates upon Wnt stimulation will potentially address the mechanisms.

Although we are currently undertaking these experiments, these by nature are time consuming and are beyond the scope of this manuscript.

3. *Figure 4E,F: The role of GSK3 in PAWS1-induced gene regulation is confusing. The reduced potential to overcome PAWS1-induced upregulation of Siamois could be explained by insufficient GSK3 expression. However, in view of their model that PAWS1 modulates β -catenin levels and activity through CK1 α and the results shown in Figure 3I, how do the authors explain that PAWS1-deficiency abrogates the ability of the GSK3 inhibitor to induce TOPFLASH activity?*

Response: Our data indicates that the levels of phospho- β -catenin at Ser⁴⁵ do not change significantly upon loss of PAWS1 expression. However, the stabilised, nuclear accumulated β -catenin appears to be substantially lower in PAWS1-KO cells compared to the WT cells. Hence, in PAWS1-KO cells, even if there is more active non-phospho β -catenin following GSK-3 inhibition, if it cannot translocate to the nucleus properly, Wnt-dependent transcription would still be subdued in PAWS1-KO cells compared to the WT.

4. *Figure 6D: Information about PAWS1 and CK1 α mRNA levels in the panel of cancer cell lines need to be provided to corroborate positively correlated expression of PAWS1 and CK1 α specifically at the protein level.*

Response: We have now included the mRNA expression data for different cell lines and the data suggests that there is no significant correlation between PAWS1 and CK1 α expression at the mRNA level (**new Fig EV3D-E**).

5. *Figure 7C-F. The data cannot be unambiguously interpreted and the regulatory relationship between PAWS1 and CK1 α remains unclear. The experimental set up in Figures 7C/F suggests that CK1 α is a negative regulator of PAWS1 independent of its kinase activity because CK1 α can neutralise the axis-inducing capacity of PAWS1. This contrasts with the remainder of the manuscript where the authors rather seem to argue that PAWS1 inhibits CK1 α to facilitate activation of Wnt signalling. To make things consistent and more informative it should be investigated how PAWS1 co-expression affects the ability of CK1 α to induce a secondary axis when CK1 α is injected at higher amounts. Likewise, it needs to be shown whether or not kinase dead CK1 α has axis inducing abilities.*

Response: Our original intention in this experiment was to show that PAWS1 and CK1 α could act synergistically to induce axis induction in the early *Xenopus* embryo. However, much to our surprise, the co-expression of the two inhibited rather than stimulated axis induction. We have replaced Fig 7F with a new panel (**new Fig 7E**) that shows the effect of co-expressing xPAWS1 with CK1 α , as well as the e, and d isoforms at higher concentrations. We find that only co-expression of the CK1 α isoform blocks xPAWS1's ability to induce a secondary axis. Co-expression of CK1e, the catalytically inactive (kinase dead) CK1e, and CKd had no effect. This in line with our interaction data, in which PAWS1 specifically binds to the CK1 α isoform but not CK1e and CKd. Moreover, in this assay the catalytically inactive (kinase dead) version of CK1 α (CK1 α KD) did not induce a secondary axis. Therefore, we would argue that CK1 α kinase activity is required for the axis induction in a complex with PAWS1, as the PAWS1 mutants that fail to bind CK1 α are unable to induce a secondary axis. Therefore, our data would suggest that CK1 α does not act as a negative regulator of PAWS1, but rather that a critical level of a PAWS1/CK1 α complex is required to induce Wnt signalling.

6. *Figure 8A: I disagree with the authors' description of the data shown. To me, there appear to be differences in total amounts of β -catenin and the dynamics of phosphoforms of β -catenin and LRP6 in PAWS1 wild-type and PAWS-deficient cells. I recommend to quantify and normalize the Western blot data to facilitate the comparison of total β -catenin, LRP6, and their phosphorylated derivatives under the different experimental conditions.*

Response: For Figure 8A (**new Figure 9A**), we quantified the blots as suggested by the reviewer (**Response Figure 1**), but these supported the overall conclusions we made. Due to lack of space allowed for Expanded View figures, we include these with this response for the Reviewer's perusal.

Response Figure 1. Densitometric quantification of the immunoblots of Figure 9A. Data are expressed as fold changes of the protein expression at each indicated time point relative to time point 0h. Quantification was performed with ImageJ software. Protein expression levels were normalised to GAPDH and the phosphorylated protein levels were normalised to the equivalent total protein levels.

We also performed similar quantifications of Western blots for Figure 8B and include them (**as new Figure 9B**). Compared to wild type and PAWS1^{WT} rescue cells, both basal and Wnt-induced levels of β -catenin-pSer⁴⁵, β -catenin-pThr⁴¹/Ser³⁷/Ser³³, active β -catenin and to some extent total β -catenin in nuclear, but not cytoplasmic, fractions were markedly lower in PAWS1^{-/-} and PAWS1^{F296A} rescue cells (Fig 8B). This was particularly evident when we analysed the levels of active β -catenin in each fraction relative to that present in the cytoplasmic fraction of WT U2OS cells (Fig 8B, lower panel), suggesting that PAWS1 potentially promotes nuclear accumulation of β -catenin.

7. *Figure 8B: In contrast to the immunofluorescence analyses shown in Figure 5F it appears that PAWS1 and CK1 α do not completely colocalize and are inversely distributed in cytosolic and nuclear fractions of wild-type cells. This differential distribution seems to be distorted upon re-expression of PAWS1 at higher than physiological levels. Furthermore, Wnt3a stimulation seems to increase cytoplasmic PAWS1 while decreasing nuclear CK1 α in wild-type cells. Again, quantification of the Western blot results should be helpful for the interpretation of the data. Possibly, results obtained under conditions of PAWS1 overexpression need to be re-evaluated concerning the potential mechanisms of CK1 α regulation.*

Response: The reviewer is correct in the assertion that the observations that the endogenous CK1 α distribution by IF in Fig 5F may be distorted by over-expression of WT PAWS1 in PAWS1-KO cells (as PAWS1 has a stabilizing effect on CK1 α protein). The baseline CK1 α IF signal in PAWS1-KO or PAWS1-F296A-rescue cells is rather weak, and these observations led us to investigate the relationship between PAWS1 and CK1 α at the protein level. In the WT U2OS cells, however, there are other FAM83 members (FAM83A-H) that can all interact with CK1 α and also determine the distribution of CK1 α , as our Supplemental manuscript (Fulcher et al, 2017) (which was included with the submission) demonstrates. In addition to PAWS1, endogenous CK1A α interacts and co-localizes with seven other members of the FAM83 family and therefore, the total CK1 α levels include those that may be in complex with other FAM83 members. With overexpression of PAWS1 in PAWS1-KO cells, there is an increase in the levels of endogenous CK1 α levels and because the increase can be attributed to directly to PAWS1 overexpression, the distribution is indeed more reflective PAWS1 levels in the two compartments. This is consistent with our hypothesis that overexpression of wild type PAWS1 modulates physiological CK1 function in such a way that leads to the activation of Wnt signalling, which the CK1-interaction deficient PAWS1-F296A mutant does not.

8. *The authors propose that the precise balance between PAWS1 and CK1 α is of critical importance for the mutual control of their stability and their modulation of Wnt signalling. In this regard the observation that PAWS1 is not the only FAM83 family member which interacts with CK1 α , could be highly relevant. This may result in a competitive setting where the consequences of PAWS1 overexpression and deficiency are nearly impossible to control and to unequivocally interpret. Functional redundancy could also be an issue. The authors should at least mention the results of the accompanying manuscript by Fulcher et al. in sufficient detail (all FAM83 family members interact and colocalize with CK1 α) and discuss the potentially confounding impact on the study presented here.*

Response: As in our response to point 7 from the reviewer, we totally agree with the reviewer. Given the numbers of CK1 substrates (for different isoforms) that have been established within the

Wnt signalling pathway, and the ability of all FAM83 members to interact with different CK1 isoforms, issues of competition, cooperation and functional redundancy are bound to be key. Our preliminary observations have suggested that in addition to PAWS1, FAM83F also induces axis duplication in *Xenopus* embryos and Wnt-transcriptional activity in cells. In contrast to PAWS1, which is localized in the cytoplasm and the nucleus, FAM83F localises primarily to the plasma membranes. Here, the mode of regulation of Wnt signalling appears to be cooperative (because of differential subcellular distribution) rather than competitive but we have yet to perform comprehensive experiments to establish the full picture. Furthermore, we have not yet established how other FAM83 members impact CK1 activity in Wnt signalling. Nonetheless, the data we present here on the role of PAWS1 on Wnt signalling through association with CK1 α is robust.

Minor issues:

9. Abstract: line 8 word duplication „that that"
10. Figure 2C: What is the prominent signal that appears in the P-SMAD3 Western blot upon BMP treatment of cells?
11. Figure 3E: The analysis of Chordin expression should be mentioned in the manuscript text, not only in the figure.
12. Figure 8: The spelling of wnt3A/WNT3A is inconsistent in panels A and B.
13. Supplemental Figure S1: There is a discrepancy between the main manuscript text and the figure legend concerning the amount of RNA injected (1 ng or 500 pg?).
14. Supplemental Figure S2: There is a mislabelling of the panels in the figure legend.
15. Supplemental Figure S2: Please adjust the column width in Figure S2 panel D.
16. Supplemental Figure S4: Please align labels of time points and lanes in panel B.

Response: We thank the reviewer for identifying these issues. We have corrected all of the minor issues raised in the manuscript.

Referee #2:

*This manuscript introduces a FAM83 family member, PAWS1, as a regulator of the Wnt pathway. The author show by both overexpression and depletion that PAWS1 has a strong influence on Wnt signalling, monitored through expression of reporter gene, of known direct target gene, as well as the ability to induce a secondary axis in *Xenopus* embryos. The phenotypes are impressive, posing PAWS1 as an important regulator of the pathway.*

The author further show that PAWS1 binds casein kinase Ialpha, and that this interaction accounts for the role of PAWS1 in Wnt signalling. These experiments, which include the use of point mutated PAWS1 variants, are extremely convincing.

The function of PAWS1 is narrowed down by showing that a) CK1a phosphorylates PAWS1, yet the phosphorylated residues are not required for PAWS1 activity; b) PAWS1 has no effect on CK1a phosphorylation activity toward a generic substrate, and c) the presence of PAWS1 has a striking stabilization effect on CK1a levels.

*Altogether, this is an impressive piece of work, combining functional experiments in *Xenopus* study with beautiful biochemistry in cell lines, using perfect controls (PAWS1 -/- cells, knockin rescues,...). The data are clean and fully convincing.*

What remains a weak aspect of the manuscript is the actual mode of action of PAWS1:

1) What is the impact of PAWS1-CK1a interaction? I am certainly aware that a complete biochemical explanation would be beyond the scope of this first study, but a simple set of IP experiments would clarify this issue right away by discriminating between two obvious alternatives: Is PAWS1 sequestering CK1a away from the Axin complex, or is the PAWS1 recruited to the Axin complex? In the first scenario, the Axin-CK1a interaction should be decreased in the presence of PAWS1. In the second scenario, both CK1a and PAWS1 should coprecipitate with Axin.

Response: As we have responded to Reviewer1-Point 2 in much greater detail above, we undertook several approaches, including the experiments that the reviewer suggested, to address potential mechanistic insights.

Briefly, we immunoprecipitated CK1 α , Axin1 and β -catenin from control and Wnt3A-treated WT or PAWS1-null U2OS cells to look for components of the destruction complex. No Axin1 and β -catenin were detected in CK1 α IPs, while as expected PAWS1 was only detected in WT but not PAWS1-KO cells, regardless of the Wnt3A-treatment (**new Fig 8**). No CK1 α or PAWS1 were detected in AXIN1 IPs regardless of the Wnt3A-treatment in any cell line. In Axin1 IPs, we were able to detect GSK3 α/β , and very low levels of β -catenin, although neither Wnt3A treatment nor PAWS1-status appeared to alter the nature of these complexes (Fig 8). Similarly, in β -catenin IPs, we were able to detect low levels of GSK-3 β , as well as CK1d and CK1e but not CK1 α . We also undertook an unbiased proteomic approach to identify interactors of PAWS1-GFP expressed in U2OS cells under unstimulated and Wnt3A (3h) stimulated conditions (**new Fig EV5B-C**). However, other than CK1 α and CK1 α -like, no major components of either the Wnt pathway or the destruction complex, including Axin1, GSK-3, APC and Dishevelled, were identified in the GFP-PAWS1 IPs under both control and Wnt3A-stimulated conditions (**new Fig EV5C**). In other experiment, we can detect CK1 α kinase activity in PAWS1 IPs from WT but not PAWS1-KO U2OS cells (**new Fig 6A**), suggesting PAWS1 does not inhibit intrinsic CK1 α kinase activity. This implies that the PAWS1-CK1 α complex potentially regulates phosphorylation of key target(s) in the Wnt signaling, at least somewhere downstream of the destruction complex and possibly factors that determine the nuclear localization of β -catenin. Although beyond the scope of the current study, we are currently undertaking comprehensive phospho-proteomic studies in WT, PAWS1-KO and PAWS1-Res cells treated with(out) Wnt3A to uncover these.

2) *No explanation for destabilization of PAWS1 and CK1 α in the absence of their partner is presented. Supplemental Fig S4 panel A, in wt cells there seem to be a stabilization of both PAWS1 and CK1 α upon inhibition of the proteasome. Wouldn't this suggests that ubiquitination/ proteasomal degradation is involved?*

Response: Indeed, the PAWS1-CK1 complex stabilization in WT cells upon proteasome inhibition implies some involvement of ubiquitin-proteasome system. However, given that in PAWS1^{-/-} cells there is no restoration of CK1 α to the levels seen in WT cells by proteasomal inhibition, the loss in CK1 α levels caused by PAWS1 loss cannot be explained by proteasomal degradation or even lysosomal degradation. Although we still do not know how the protein levels of CK1 α is reduced in PAWS1^{-/-} cells, we postulate co-translational assembly and stabilization as a possibility and will address this in the future.

3) *The authors propose that the right levels of PAWS1 and CK1 α are required for proper regulation of Wnt signalling. This hypothesis is based in particular on the analysis of induction of secondary axis in Xenopus. Page 9: "human CK1 α can also induce a secondary axis in a dose dependent manner in Xenopus embryos (Fig 7C). However, when co-expressed with PAWS1, the axis induction was blocked (Fig 7D)." Yet, this statement is not supported by the data, because the dose of CK1 α mRNA used in 7D (50ng) is not sufficient alone to induce secondary axis. The experiment should be performed with the high CK1 α dose (200ng). Panel F: Doses for PAWS1 and CK1 α kinase dead mRNA not indicated.*

Response: We have now done this experiment with doses of CK1 α mRNA that alone are sufficient to induce a secondary axis (**new Fig 7E**). Again, we find that co-expression of CK1 α and xPAWS1 abrogates the axis duplication phenotype. We have now updated the figure legend with the amounts of each mRNA used.

Further issues:

4) *Related to point 1. PAWS1 has no effect on CK1 α activity on a peptide substrate, but could have an effect on endogenous substrates e.g. within the Axin complex. Please comment.*

Response: We agree with the reviewer that the association of PAWS1 with CK1 α could indeed affect access of a bona fide protein substrate to CK1 α in cells. However, what the in vitro CK1 α assays, as well as the kinase assays on PAWS1 IPs from WT cells (**new Fig 6A&B**), do show is that PAWS1 does not inhibit the intrinsic CK1 kinase activity in vitro. This has led us to postulate that

the PAWS1-CK1 α complex determines a subset of context-specific CK1 α substrates in specific subcellular compartments. We are currently performing comprehensive phospho-proteomic experiments in WT, PAWS1^{-/-} and PAWS^{WT} rescue cells with and without Wnt3A stimulation to uncover potential PAWS1-dependent CK1 α substrates in the Wnt signalling pathway.

5) Fig8B and bottom of page 10: A kit is used to separate cytoplasmic and nuclear fractions, from which conclusions are drawn about the effect of PAWS1 on cytoplasmic/nuclear β -catenin. However, most β -catenin is supposed to be present at the plasma membrane. In which fraction are membranes recovered? Please include a plasma membrane marker (cadherin, LRP6 or other) and adjust interpretation according to result.

Response: We thank the reviewer for the suggestion. We used a different kit that allowed us to probe plasma membrane and cytoskeletal fractions, in addition to nuclear and cytoplasmic fractions. As noted by the reviewer, the vast majority of total β -catenin was detected in the membrane fraction in WT cells and Wnt3A stimulation did not significantly alter this distribution (**new Fig EV5E**). Analogous membrane distribution of total β -catenin was observed in PAWS1^{-/-} cells, and this was not altered by Wnt3A treatment (**new Fig EV5E**). However, the only slight differences that we detected between WT and PAWS1^{-/-} cells again was the apparent Nuclear/Cytoplasmic ratio in levels of total β -catenin in Wnt3A treated cells, with an increased cytoplasmic accumulation of β -catenin seen in PAWS1^{-/-} cells compared to WT cells (**new Fig EV5E**).

6) Role of PAWS1 is BMP signalling: The data presented here argue that PAWS1 does not influence this pathway, yet such role was previously proposed by the authors. I would minimally expect a short explanation in the discussion.

Response: Previously we published observations that PAWS1 modulates a subset of SMAD4-independent genes (non-canonical) in the BMP signalling pathway, but not the canonical SMAD4-dependent BMP pathway (Vogt et al, 2014). We now mention this in the Results and Discussion sections more clearly. Here, because the inhibition of the canonical BMP pathway is implicated in the body axis formation during embryogenesis, we looked at the effects of PAWS1 on canonical BMP signalling in Xenopus and U2OS cells, which is not affected. This is consistent with our previous findings of the role of PAWS1 on BMP signalling. For clarity, we now also elaborate this further in the **Discussion section**.

Minor comments

Fig3A. The specificity of the "active β -cat" antibody is not well established. In fact, in figure 8B, patterns for "active" and total β -cat may be identical, except for the difference in strength of the signal. I suggest to interpret these data with caution.

Response: We clarify what the antibody was raised against (non-phospho Thr41, Ser45 region) and cite a relevant reference in the Results section. Because the active β -catenin is the only antibody that we could find that appears to recognize enriched (possibly non-phosphorylated) β -catenin levels in cytoplasmic and nuclear fractions upon Wnt3A stimulation, we decided to use it.

Fig5F. PAWS1-CK1 colocalization is not quite obvious: Including a high magnification and pointing to colocalizing/non-colocalizing spots would help. One would also like to see a comparison of the degree of colocalization with another unrelated cytoplasmic protein, e.g. PAWS1/GAPDH.

Response: We now include more representative images and additional controls along with this response (**Response Figure 2 below**). For comparison, we transfected U2OS cells with GFP-SMAD3, which does not pull down CK1 α (from our previous studies), and performed anti-CK1 α IF. Although both GFP-SMAD3 and CK1 α appear in nuclear compartments, there is a minimal overlapping fluorescence (**Response Figure 2**). On the other hand, we refer to co-localization data with other FAM83 members and CK1 isoforms in the attached Supplemental manuscript (Fulcher et al, 2017), which display much clearly the co-localization patterns in discreet subcellular compartments (e.g. see FAM83C, FAM83F and FAM83H), and display selectivity between α , δ and ϵ isoforms (Fulcher et al, 2017).

Fig6C: In CK1a KD, PAWS1 is clearly decreased, but still half is left, which implies that PAWS1 can subsist in the absence of CK1a. Please comment.

Response: There are two possible explanations for this. First, PAWS1 also interacts with CK1 α -like isoform (encoded by CSNK1A1L gene), which is poorly characterised, and its expression could account for remaining PAWS1 stability (**new Fig EV5C**). Second, the rate of PAWS1 turnover could be longer than that of CK1 α , which would account for incomplete loss of PAWS1. Additional possibilities we cannot completely rule out include: i. incomplete knockdown of CK1 α (limited perhaps by the sensitivity of the CK1 α antibody employed for Western blot) and ii. possibility that not all of PAWS1 exists in complex with CK1 α .

Fig S2E: How come CK1a is not visible in input?

Response: This was due to adjustment of IB exposure to allow the unsaturated detection of the relatively higher levels of CK1 α in IPs compared to inputs. We have now also included a higher exposure of the blot, where CK1 α is visible in the inputs (**new Fig EV10**).

Legend Fig S2: Correct reference to panels (a,b,b,c,d)

Response: Now corrected (**new Fig EV1**)

FigS3E: PAWS1 phosphorylation: not required for ectopic axis duplication. But could be important for endogenous regulation?

Response: Obviously, this is a possibility but currently we lack robust phospho-specific antibody to allow detection of the endogenous Ser614 phosphorylation. Furthermore, we were unable to detect pSer614 definitively in IPs of endogenous GFP-PAWS1 from unstimulated or Wnt3A stimulated cells, under conditions where some other PAWS1 phospho-peptides were detected (**new Figure EV5D**). We are generating an antibody against pSer614 to investigate its potential role further.

FigS4B: what is LC3?

Response: The assessment of LC3 (Microtubule-associated protein 1A/1B-light chain 3; encoded by the MAP1LC3A gene) flux is a well-established method for visualising autophagic activity. LC3 exists in lipidated (LC3-II) and non-lipidated (LC3-I) forms, which display distinct mobility patterns upon electrophoresis. Lipidation of LC3 is a prerequisite for autophagy to occur, as it becomes conjugated to the autophagosomal-bound lipid phosphatidylethanolamine in response to autophagy-inducing stimuli, such as amino acid starvation. As autophagic cargo is quickly destroyed upon fusion with the lysosome, the visualisation of the LC3 flux through the autophagic pathway is aided by the use of the lysosomal fusion inhibitor bafilomycin A1. We now provide sufficient information in the Figure legends and citations in the Results section.

Referee #3:

The manuscript from Bozatzki et al describes a role for PAWS1/FAM83G in regulation of Wnt signaling. In Xenopus embryos and U2OS cells, over-expression of PAWS1 activates β -catenin signaling, while knockout inhibits signaling. In search of mechanism, they find strong interaction with CK1 α , a known regulator of Wnt signaling. Epistasis studies place PAWS1 at or near the destruction complex. Point mutations in PAWS1 that block interaction with CK1 α also block the Wnt/ β -catenin stimulating activity of PAWS1. The data taken together suggest that PAWS1 acts like a CK1 α sink that can sequester active CK1 away from relevant targets in the Wnt pathway. However, here there is no clear evidence for changes in the phosphorylation of the myriad CK1 targets. Whether PAWS affects other CK1 dependent processes is not well addressed. The quality of the data on whole is good, with specific exceptions noted below. The mechanism of PAWS1 activation remains unsettled. The paper does not settle on mechanism, which will reduce its impact in the field. It is also odd, and not particularly well addressed, that this must not be a core mechanism in Wnt signaling since other Wnt-responsive cell lines do not expression PAWS.

A few key questions remain.

Key experiments I would like to see:

1. They show that active or inactive CK1 α reverses the signaling effect of PAWS1. Does CK1 δ or CK1 ϵ expression also reverse this signaling effect?

Response: This is a very interesting question. Consistent with our interaction data that shows that PAWS1 interacts with only the CK1 α isoform, we find that co-expression of either CK1 ϵ , the kinase dead CK1 ϵ or CK1 δ with xPAWS1 has no effect on xPAWS1's ability to induce axis duplication. We have now combined this data with the catalytically inactive CK1 α KD data into a single panel (**new Figure 7E**).

2. Does PAWS1 expression change the amount of CK1 isoforms co-immunoprecipitating with axin or Dishevelled? That might be a simple experiment to see if PAWS1 simply removes CK1 α from the destruction complex leading to stabilization of β -catenin.

Response: We thank the reviewer for raising these points. We have performed elaborate experiments to address these issues and provide details in **response to Reviewer 1, Point 2** above.

3. Does PAWS expression alter other CK1 dependent processes? This is not yet well addressed.

Response: We will address this when we undertake comprehensive phospho-proteomic analysis on PAWS1-dependent CK1 α substrates within and beyond Wnt signalling pathways. Perhaps also noteworthy here are our findings that the FAM83 family of proteins anchor different CK1 isoforms to different subcellular compartments (Fulcher et al, 2017; included as a Supplemental manuscript with this submission) in much the same way as PAWS1 and this collective regulation of CK1 isoforms potentially streamlines CK1 substrates in different biological contexts.

General comment: plunger plots are thankfully going out of fashion and should not be used here. Experiments with small samples sizes should use scatter-plots, dot plots or similar methods that permit direct evaluation of the distribution of the data. C.f. Weissgerber et al. (2015) Beyond Bar and Line Graphs: Time for a New Data Presentation Paradigm. PLoS Biol 13(4): e1002128. doi:10.1371/journal.pbio.1002128

Response: We have replaced all the plunger plots with scatter plots. We thank the reviewer for excellent suggestion – this is such a great way of presenting data! For visual separation of different conditions (e.g. Wnt3A treated vs untreated), on top of scatter plots, we have included lightly shaded bars.

Other questions and issues:

As a comment, CK1 plays multiple roles in Wnt signaling including phosphorylation of β -catenin, Disheveled, LRP6, LEF1 and APC. These papers should be referenced and considered in the discussion.

Response: We thank the reviewer for the suggestions. We have now updated citations with relevant references in the Discussion section.

Fig 1C how much hPAWS mRNA was injected?

Response: the amount of mRNA has now been added to the figure legends.

Fig 1E&F n=3 seems like a very small n to have much confidence in the results.

Response: We have increased the n in this experiment and indicated the changes. The data is now expressed as a fold change relative to expression in whole embryos. We have also supplemented the qPCR analysis with an additional panel to the figure (**new Fig 1E**) that clearly shows by in situ hybridisation the dorsalisation of ventral blastomeres following ectopic xPAWS1 expression. Chordin expression was induced in VMZ and expanded in the DMZ when xPAWS1 mRNA was expressed in ventral and dorsal blastomeres respectively. Further, we show that Vent2 expression is repressed in the VMZ following ectopic xPAWS1 expression.

Fig 3A needs molecular weight markers. I'm confused by what Fig 3B and C are quantitating. It would help to specify which antibody is used in each case. The quantitation in fig 3C doesn't seem to match with what I can see in the "active β -catenin" blot of Fig 3A.

Response: We have added the molecular weight markers, and have amended the figure legend to more clearly define the antibodies that were used.

We have also amended the figure legend to more clearly state what was quantitated in the Western blot. In previous Panel 3B (**new Fig 3C top panel**), we are quantitating the level of stabilized b-catenin_GFP that accumulated in the presence or absence of xPAWS1_mCherryHA expression using an anti-GFP antibody. We normalized b-catenin_GFP in the blot to a-tubulin. We then quantitated the extent of xPAWS1-induced b-catenin stabilization by the fold change in the levels of b-catenin_GFP in xPAWS1_mCherryHA embryos relative to its expression in embryos injected with b-catenin_GFP alone. Therefore, there was a 10-fold increase in stable b-catenin_GFP in the presence of xPAWS1. The second analysis (**new Fig 3C bottom panel**) compared the level of 'active' unphosphorylated b-catenin_GFP in the presence or absence of xPAWS1_mCherryHA. This was done in a similar manner, first by normalizing to a-tubulin and then expressing the fold change of 'active' b-catenin_GFP in double injected embryos relative to the levels in embryos injected with b-catenin_GFP alone.

It is surprising in Fig 3A: why does xPAWS stabilize β -catenin_GFP but has no noticeable effect on endogenous "active" β -catenin? Would it be helpful to also probe for total β -catenin?

Response: The reason that we did not observe an effect on endogenous 'active' b-catenin is due to our experimental design. We originally designed this experiment using an epitope tagged b-catenin in order to distinguish the fraction of both stabilised and unphosphorylated 'active' b-catenin following extraction of a whole stage 10 embryo. At this stage b-catenin signalling is active and therefore contains large pools of 'active' unphosphorylated b-catenin. We reasoned that as we were extracting protein from the whole embryo any increase in active b-catenin due to PAWS1 activity may be masked by these large pools of endogenous protein. We also injected the tagged b-catenin mRNA into the animal pole, away from the endogenous active wnt-signalling centre in the DMZ,

such that any stabilized b-catenin_GFP that did accumulate would be due solely to the activity of xPAWS1 protein.

This is evident in **Fig 3A**: the level of xPAWS1-dependent active b-catenin_GFP is very small compared to the level of endogenous active b-catenin present in the whole embryo at stage 10 (Fig 3A middle panel: compare the low intensity of the active b-catenin_GFP (upper red band, highlighted with a single white asterisk) vs the high intensity band representing the endogenous active b-catenin (lower red band; double white asterisk).

However, we agree with the reviewer that it would be informative to probe for total endogenous and 'active' b-catenin following xPAWS1 expression. Therefore, we injected embryos with xPAWS1 mRNA into the animal pole at the 1 cell-stage and examined the effects of PAWS1 on the pools of total and active b-catenin in isolated animal caps at stage 10. By using naïve animal caps rather than the full embryo as we did in Fig 3A, we could exclude the large pools of b-catenin present in the DMZ that would mask the effects on b-catenin due to xPAWS1 activity. New **Fig 3B** now shows the levels of both 'active' and total b-catenin present in naïve animal caps in the presence or absence of xPAWS1. We see a 2-fold increase in total b-catenin (new **Fig 3B**, lower panel, quantitation in new **Fig 3D** top) following the injection of xPAWS1 mRNA and just over a 3-fold increase in active b-catenin compared to uninjected caps. (**Fig 3B** upper panel, quantitation in **Fig 3D** bottom).

Fig 3D: the figure legend states "Wnt activity, stimulated with 50 μ M of the GSK3-b inhibitor CHIR99021, induces stabilization and nuclear localization of β -catenin_GFP in the absence of xPAWS1 expression." However, that is not visible in the figure, second row. Unless the authors are referring to some faint nuclear signal very unlike that seen with xPAWS1 co-expression. This requires clarification.

Response: We agree with the reviewer that it is hard to see any stabilized b-catenin in these panels. We have removed both of the CHIR99021 panels and now refer the reader to **new Fig EV1D-I**. The premise in the experiment was to inject b-catenin-GFP at levels low enough that any b-catenin GFP that accumulated would be due solely to PAWS1-dependent activity and not due to excess tagged-GFP protein overloading the endogenous destruction complex (in our hands, doses of b-catenin_GFP mRNA in the 200-250pg range produces detectable stable protein and induces a secondary axis). However, as a control we wished to show that our control cells do in fact express b-catenin_GFP mRNA and that a short 3 hr treatment with the GSK-3b inhibitor CHIR could stabilize a small proportion of b-catenin_GFP in the absence of xPAWS1. This is what is shown in **new Fig EV1D-I**. In contrast, as the reviewer notes, the response due to xPAWS mRNA is robust. In this case, we are looking at the effect of translated xPAWS1 protein that has accumulated since the time the mRNA was injected at the 1-cell stage. This equates to over 20 hrs prior to our analysis by either Western blotting or by confocal imaging.

The figure legend is unclear - is figure 3E done with tagged or untagged xPAWS? How much mRNA of each construct was injected?

Response: We injected a Myc-tagged (MT) version of xPAWS1 in these experiments. We have now updated the figure legend with the doses of mRNA for both constructs.

Fig 3F-H: is it possible that failure to induce axis and Siamese expression is due to production of an unstable protein? Without evidence of protein expression one cannot conclude much from negative results in DUF, 151, etc... mutants.

Response: We have added a panel (J) to **new Fig EV1** with a Western blot showing the expression of each of the Myc-tagged(MT)-xPAWS1 mutants.

Odd that PAWS doesn't do nearly so much in HEK293 cells as it does in U2OS. This is consistent with the data in fig 6D, where PAWS1 is low or undetectable in a number of cell lines that presumably are responsive, like HEK293 cells, to Wnt ligands. It is known that other CK1 genes (CK1 δ , CK1 ϵ) can also prime β -catenin for destruction. Have the authors probed the same blots for CK1 δ and CK1 ϵ ?

Response: Clearly cellular context, in which the relative expression levels of factors (including feedback factors) that either positively or negatively regulate Wnt signalling, is key when comparing different cell lines with regard to the extent of Wnt responses. What was striking from Fig 6D (**now Fig 6E**), regardless of the potency of Wnt signalling in each cell line, was the correlation between PAWS1 and CK1 α protein expression. The reason we probed only CK1 α here is because our MS analysis and Western blots indicated that endogenous PAWS1 interacts only with CK1 α but not CK1 δ or CK1 ϵ (the specificity data is included in Fulcher et al, 2017 included as relevant Supplemental manuscript with original submission). We now include a blot for CK1 ϵ , and find that CK1 ϵ protein expression does not correlate with PAWS1 expression (**new Figure 6E**). However, we still need to be cautious though, because all 8 FAM83 members (A-H) can interact with CK1 α and half of them (A,B,E&H) also interact with CK1 δ and CK1 ϵ (Fulcher et al, 2017; which was included with the original submission), so any correlation between expression probably depends on the relative expression levels of individual FAM83 members and their relative affinities for each CK1 isoform.

In these types of signaling assays, the amount of plasmid transfected is important to note. Please include this information, preferably in the figure legends.

Response: For transient transfections in Wnt Luciferase reporter assays, the Methods section indicates the amounts of relevant plasmids used. If the reviewer has any other specific examples where clarity is lacking, we will be happy to elaborate and clarify.

In Fig 4, the loss of signaling in U2OS cells knocked out for PAWS is impressive. It would be important to test if this effect was specific to the Wnt pathway, e.g. by testing other signaling pathways to test if they were similarly affected. Clearly, some cells without PAWS (HEK293 cells) are highly responsive to Wnt signaling. The NF-AT experiments in S5 are not compelling (see note below).

Response: Here we show that canonical BMP signalling pathway is not affected by overexpressing or knocking out PAWS1 from embryos and cells. Similarly, in PAWS1-KO U2OS cells, TGF β -induced SMAD2/3 phosphorylation is not affected compared to WT cells. While we had biochemical and phenotypic rationale for testing the effects of PAWS1 on BMP, TGF β and Wnt signalling, insights into potential analyses on PAWS1 function on other signalling pathways are limited – hence we will consider these in the future. Our proposed phospho-proteomic screens which will potentially identify PAWS1-dependent CK1 α substrates will hopefully guide us on specific signalling processes and physiological responses that PAWS1 may regulate.

In figure 5D, it appears there is near complete depletion of CK1 α in the flowthrough after expression of WT PAWS1. That is impressive. It would be interesting (not essential) to know the fold overexpression of PAWS1 above endogenous.

Response: We have included anti-PAWS1 blot to visualise the endogenous PAWS1 levels.

Fig 7A and 7B are strong evidence for the importance of the PAWS1-CK1 α interaction. Fig 7B: please clarify in text or figure legend, these are transient transfections?

Response: These are cell lines stably restored with retroviruses encoding PAWS1 as shown in new Fig 6C. This is clarified in the figure legends.

I find fig 8 both difficult to interpret, and not illuminating. First, it's unclear if this was a one-off, or a multiply replicated experiment with statistical confidence in the small changes. Second, while it seems to confirm their prior data on β -catenin regulation, it doesn't help me understand mechanism. I don't think this adds tremendously to my understanding.

Response: We have included quantification of the Western blots for a clearer interpretation of Figure 8B (**new Fig 8B**). What we want to highlight is that the levels of the nuclear β -catenin change in the PAWS1 KO cells versus the wild type and the rescue cells. For Figure 8A, we quantified the blots as suggested by the reviewer, but these supported the overall conclusions we made. Due to lack of space allowed for Expanded View figures, we include these with this response (**Response Figure 1 in response to Reviewer 1, point 6 above**) for the Reviewer's perusal.

Minor: please include molecular weight indicators on all SDS-PAGE gels and immunoblots.

Response: These are now included.

Supplemental figure 2D and E legends need to be corrected.

Response: These are now corrected and included in **new Fig EV1**.

Supplemental figure 2E: Why is no CK1 α present in the input? It's visible in U2OS cells in many other figures. It's important because the absence of CK1 α in the flow through might have suggested complete depletion of CK1 α , but not if we can't see it in the input.

Response: We have included a higher exposure blot where CK1 α is visible. We ran the inputs and the immunoprecipitations on the same gel and due to the high enrichment in the latter, a high exposure saturates the signal (**new Fig EV10**).

The Mass spectrometry top hits should all be listed in a table in supplemental data so that we can assess if there were other high probability interactors.

Response: We now include a graph representing the top interactors (**new Fig EV5C**). We also make note of interactors that were common for both endogenous PAWS1^{GFP/GFP} and overexpressed PAWS1-GFP. These data will be deposited in public database (PRIDE) once the paper is published.

Supplemental Fig 5 lack a control for CK1 activity. If the model is, increased CK1 activity drives NFAT to the nucleus, they need to show this happens in their system, either by expression of CK1, or inhibition of CK1 in the presence of the ionophore. Otherwise we can't interpret this negative result. I also wonder if the dose of ionophore is too high, since there looks to be a lot of apoptosis in the samples?

Response: The model is based on a previous report (Okamura et al., 2004, Mol Cell Biol., **24**:4184), which showed that CK1-dependent phosphorylation of key serine residues within NFAT prevents NFAT nuclear accumulation. When calcineurin, a calcium and calmodulin dependent phosphatase, is activated, it can target NFAT for dephosphorylation, thereby triggering NFAT nuclear translocation.

To show that CK1-activity is important for cytoplasmic retention in our assay we have injected embryos with a catalytically inactive (kinase dead) CK1a construct (CK1a KD). Similar to what was has been reported by Okamura et al, we find that inhibiting CK1 activity re-distributes NFAT_GFP to both the nucleus and the cytoplasm. New **Fig EV4C** includes panels to show nuclear localization of NFAT_GFP in animal cap cells expressing NFAT_GFP alone, NFAT_GFP + MTxPAWS1, and NFAT_GFP + CK1-a KD.

These cells are healthy and are actively dividing. Xenopus animal cap cells plated on cadherin coated coverslips have lobular nuclei (not uniform as seen in standard mammalian cell lines). This is evident in the higher magnification images provided (nuclei labelled with Hisone 2B_RFP (H2B_RFP)). The Ca⁺² ionophore treatment was a control we included to show that the NFAT_GFP protein remained under the control of the Ca⁺² signalling pathway. This was the endpoint of our assay (i.e., images are 10 min post Ca⁺² ionophore treatment). Under these culture conditions, NFAT-GFP rapidly translocates to the nucleus, yet the cells are not overtly affected by Ca⁺² ionophore treatment. We have provided a 1 hr movie of cells co-expressing NFAT_GFP and H2B_RFP mRNAs treated with Ca⁺² ionophore (time of Ca⁺² ionophore addition @ 4 min; each frame was captured every 2 minutes for 60 minutes) to show that cells continue to behave normally. This is included below with the responses for the Reviewers' perusal.

Response Movie 1 (Movie not shown): A 60 min lapse of the behaviour of animal caps cells following Ca⁺² ionophore treatment. Dissociated animal caps expressing NFAT_GFP and H2B_RFP mRNAs were cultured on cadherin coated glass slide and then treated with Ca⁺² ionophore (25 μ g/ml) at t=4 min. Each frame was taken at 2 min intervals for a total of 60 min. For

the duration, cells clearly continue to behave normally, suggesting no overt toxicity by the Ca²⁺ ionophore.

2nd Editorial Decision

8 December 2017

Thank you for the submission of your revised manuscript. I am sorry for the slight delay in getting back to you; we have now received the comments from all referees, which are pasted below.

As you will see, all referees acknowledge that the study has been improved and all overall support the publication of your work. However, referee 1 also makes 2 important comments that I think should be addressed experimentally. Please let me know if you have any issues with these last two requests so that we can discuss this further.

Figure 4A, 6B, 7B, EV3F and EV4B do not specify "n" and the error bars, please add this information to the figure legends.

I am looking forward to receiving a newly revised manuscript as soon as possible.

REFEREE REPORTS

Referee #1:

Comments on revised version of manuscript number: EMBOR-2017-44807-T: "PAWS1 controls Wnt signalling through association with Casein Kinase 1 α " by Bozatzi/Dingwell and colleagues.

In the revised version of their manuscript the authors have undertaken major efforts to resolve the main unanswered question of their study, namely to clarify the relationship between the PAWS1::CK1 α complex and components of the β -catenin destruction machinery. In this regard they performed additional co-immunoprecipitation experiments and further explored their mass spectrometry data. The results of these additional investigations argue against the possibility that PAWS1 associates with the destruction complex and controls CK1 α activity in this context. Although the authors claim that there are no previous reports of an association of CK1 α with components of the destruction complex, there is at least one prior report of such an interaction (Li et al., Cell 149, 1245-1256, 2012 [Fig S4, Table S1]). It would be futile to speculate about the reasons for the this discrepancy and the negative results of Bozatzi/Dingwell and colleagues, but at least in their system and under their experimental conditions the authors can rule out one model for PAWS1 function, and I am satisfied by the results of the additional experimentation. In addition, the authors nicely responded to almost every other of my major and minor concerns. Two critical issues remain, though:

1. To allow for more convincing interpretation of the Western blots shown in Figures 9A and 9B I asked for quantification of the data. I would have preferred to see quantifications of several experiments to be able to assess reproducibility and robustness of the observations. The fact that there are fluctuations and discrepancies within the data shown in Figure 9 (e. g. active β -catenin, lanes 5 and 13 should be similar), and between the results in Figure 9B and EV5E (cytoplasmic/nuclear distribution of PAWS1 in control wild-type cells) in my opinion reinforce the need for quantification of several experiments to allow for unambiguous conclusions. Likewise, Response Figure 1 confirms my suspicion that levels of total β -catenin are reduced while relative S45 phosphorylation is up in PAWS1^{-/-} cells. This contrasts with the authors' claim that "levels of phospho- β -catenin at Ser45 do not change significantly upon loss of PAWS1 expression" (quote from Rebuttal letter; also Discussion page 15, lines 7-8). Of course, the significance of the observed differences between PAWS1 wild-type and ^{-/-} cells would have to be determined by evaluation of several experiments. What is more important, an increase in relative S45 phosphorylation concomitant with reduced overall amounts of β -catenin in the absence of PAWS1 has implications for mechanistic interpretations. This hints that the function of PAWS1 could be to limit CK1 α activity towards β -catenin. Thereby, the observed effects of PAWS1 gain-of-function and loss-of-function on β -catenin intracellular localization and Wnt/ β -catenin pathway activity could easily be

explained as a result of changes in β -catenin protein turnover. Accordingly, I repeat my request to quantify Western blot data from several independent experiments, especially with regards to levels of β -catenin Ser45 phosphorylation and total β -catenin. In addition, I recommend to measure β -catenin half-life in the presence and absence of PAWS1.

3. The explanation for the failure of the GSK3 inhibitor to elicit a reporter gene response in the absence of PAWS1 is not convincing. First, β -catenin with a single amino acid substitution in the GSK3 phosphorylation site shows nuclear translocation and transcriptional activity (e. g. Kelly et al., 2011, *Cell Stem Cell* 8, 214-227), arguing that there is no regulated step once GSK3-mediated phosphorylation of β -catenin has been abrogated. Second, Figure 3J argues that PAWS1 is a limiting factor in Wnt3a-stimulated cells but not in LiCl-treated cells. Thus, GSK3 inhibition renders PAWS1 expendable, and PAWS1 appears to act upstream of GSK3. This view is in agreement with the PAWS1 effect on Ser45 phosphorylation. It is also consistent with the ability of GSK3 to block xPAWS1-induced Siamois expression as shown in Figure 4D, i.e. PAWS1 gain-of-function can be blocked by GSK3 overexpression (thus compensating for perturbation in CK1 α function). Accordingly, inhibition of GSK3 in PAWS1^{-/-} cells should lead to reporter gene activation. Since the experiments in Figures 3J and 4D involve different experimental conditions and show vastly different TOPflash responses to GSK3 inhibition I suggest to repeat the experiments shown in Figure 4E including LiCl and 12 h treatment time.

Minor issues:

1. Page 10, line 26: Please refer to Fig 7E when describing the effects of PAWS1/CK1 α KD co-expression.
2. Fig 7E: I think the second column should be labelled with "CK1 α " (wildtype not KD).

Referee #2:

This is a very interesting and well performed work.
The authors have in my view satisfactorily addressed all the comments of the reviewers.

Referee #3:

The authors have conscientiously and thoroughly addressed the issues raised. This manuscript makes a significant contribution.

Responses to reviewer's comments: The reviewer's comments are *italicized* and our responses appear as non-italicized fonts. New data and figures are indicated with **bold face fonts**.

Referee #1:

In the revised version of their manuscript the authors have undertaken major efforts to resolve the main unanswered question of their study, namely to clarify the relationship between the PAWS1::CK1 α complex and components of the β -catenin destruction machinery. In this regard they performed additional co-immunoprecipitation experiments and further explored their mass spectrometry data. The results of these additional investigations argue against the possibility that PAWS1 associates with the destruction complex and controls CK1 α activity in this context. Although the authors claim that there are no previous reports of an association of CK1 α with components of the destruction complex, there is at least one prior report of such an interaction (Li et al., Cell 149, 1245-1256, 2012 [Fig S4, Table S1]). It would be futile to speculate about the reasons for this discrepancy and the negative results of Bozatzi/Dingwell and colleagues, but at least in their system and under their experimental conditions the authors can rule out one model for PAWS1 function, and I am satisfied by the results of the additional experimentation. In addition, the authors nicely responded to almost every other of my major and minor concerns.

Response: We thank the reviewer for his/her comments and a thorough examination of our work. The reviewer points to the study by Li et al (2012, Cell, Fig. S4) to suggest that CK1 α has been shown to interact with the components of the destruction complex. However, this study used overexpressed hAxin1-Flag to demonstrate its interaction with the CK1 α . Indeed, it is based on this study that we had already contacted V. Li (Crick Institute, first author of the above manuscript) for reagents and advice on IP/interaction assays at the endogenous level in our original manuscript. Under conditions where PAWS1 and CK1 α interact robustly, we were unable to detect interactions between endogenous CK1 α and Axin1 or β -catenin in U2OS extracts. To our knowledge, all the studies that have shown CK1 α interaction with the destruction complex components are based on over-expression of one or more components. However, we emphasise that the lack of detectable interaction between CK1 α and the components of the destruction complex at the endogenous levels does not rule out the possibility that CK1 α phosphorylates one or more of the components, as by nature enzyme-substrate relationships are transient. Furthermore, in addition to PAWS1, we have shown that 7 other FAM83 proteins can also interact with CK1 α (Fulcher et al, included) and one (or more) of these could control the CK1 α -mediated phosphorylation of β -catenin or other components of the destruction complex. Although we have not tested it in this manuscript, both CK1 α and FAM83B were identified in GFP-APC and GFP-Axin (both overexpressed) IPs from Wnt-stimulated but not unstimulated extracts in a SILAC-based proteomic study (Hilger and Mann, J. Proteome Res., 2012, 11 (2), pp 982–994).

Two critical issues remain, though:

1. To allow for more convincing interpretation of the Western blots shown in Figures 9A and 9B I asked for quantification of the data. I would have preferred to see

quantifications of several experiments to be able to assess reproducibility and robustness of the observations. The fact that there are fluctuations and discrepancies within the data shown in Figure 9 (e. g. active β -catenin, lanes 5 and 13 should be similar), and between the results in Figure 9B and EV5E (cytoplasmic/nuclear distribution of PAWS1 in control wild-type cells) in my opinion reinforce the need for quantification of several experiments to allow for unambiguous conclusions. Likewise, Response Figure 1 confirms my suspicion that levels of total β -catenin are reduced while relative S45 phosphorylation is up in PAWS1^{-/-} cells. This contrasts with the authors' claim that "levels of phospho- β -catenin at Ser45 do not change significantly upon loss of PAWS1 expression" (quote from Rebuttal letter; also Discussion page 15, lines 7-8). Of course, the significance of the observed differences between PAWS1 wild-type and ^{-/-} cells would have to be determined by evaluation of several experiments. What is more important, an increase in relative S45 phosphorylation concomitant with reduced overall amounts of β -catenin in the absence of PAWS1 has implications for mechanistic interpretations. This hints that the function of PAWS1 could be to limit CK1 α activity towards β -catenin. Thereby, the observed effects of PAWS1 gain-of-function and loss-of-function on β -catenin intracellular localization and Wnt/ β -catenin pathway activity could easily be explained as a result of changes in β -catenin protein turnover. Accordingly, I repeat my request to quantify Western blot data from several independent experiments, especially with regards to levels of β -catenin Ser45 phosphorylation and total β -catenin. In addition, I recommend to measure β -catenin half-life in the presence and absence of PAWS1.

Response: We thank the reviewer for a thorough analysis of our data on Figure 9A and for suggestions that PAWS1 might act to disrupt the phosphorylation β -catenin at Ser45 by CK1 α . We have now repeated the experiments comprehensively (also by using LiCor detection method to give us actual photon counts rather than solely rely on Image J quantifications we had done previously) to confidently ascertain whether this could be a possibility. However, as the new repeats (**Response Figure 1A-C** below) demonstrate, there are no significant differences in the ratio of the phospho-Ser45- β -catenin over total levels between the WT and PAWS1^{-/-} U2OS cells over the course of Wnt3A stimulation. Therefore, it is unlikely that attenuation of Wnt signalling in PAWS1^{-/-} cells can be attributed to the phosphorylation status of β -catenin controlled by PAWS1. The phosphorylation of β -catenin at Ser45 in both the WT and PAWS1^{-/-} U2OS cells goes down at 30 min after Wnt3A-stimulation and is restored to basal levels by 6 h.

In similar vein, in order to apply statistical tests for Figure 9B, we repeated the experiments of cytoplasmic/nuclear distribution of total and active β -catenin in WT and PAWS1^{-/-} U2OS cells (**Response Figure 2**). The nuclear accumulation of the active β -catenin in response to Wnt3A stimulation is significantly attenuated in PAWS1^{-/-} cells. This is in agreement with our data from Fig 3E showing that overexpression of PAWS1 enhances the nuclear accumulation of β -catenin in *Xenopus* animal caps.

As far as the protein levels of total β -catenin are concerned, in addition to the data presented in Figure 9, in many other experiments where we have compared wild type to PAWS1^{-/-} U2OS cells upon Wnt stimulation over different time courses, we have not

observed substantial differences in total β -catenin levels. As one of the reviewers pointed out in the first round of reviews, and we showed in Figure EV5E, the majority of total β -catenin appears to be associated with the plasma membrane and this pool might not be involved in Wnt signalling. This makes any interpretation based on either negligible or small changes in total β -catenin levels when comparing WT and PAWS1^{-/-} U2OS cells more complicated.

3. The explanation for the failure of the GSK3 inhibitor to elicit a reporter gene response in the absence of PAWS1 is not convincing. First, β -catenin with a single amino acid substitution in the GSK3 phosphorylation site shows nuclear translocation and transcriptional activity (e. g. Kelly et al., 2011, Cell Stem Cell 8, 214-227), arguing that there is no regulated step once GSK3-mediated phosphorylation of β -catenin has been abrogated. Second, Figure 3J argues that PAWS1 is a limiting factor in Wnt3a-stimulated cells but not in LiCl-treated cells. Thus, GSK3 inhibition renders PAWS1 expendable, and PAWS1 appears to act upstream of GSK3. This view is in agreement with the PAWS1 effect on Ser45 phosphorylation. It is also consistent with the ability of GSK3 to block xPAWS1-induced Siamese expression as shown in Figure 4D, i.e. PAWS1 gain-of-function can be blocked by GSK3 overexpression (thus compensating for perturbation in CK1 α function). Accordingly, inhibition of GSK3 in PAWS1^{-/-} cells should lead to reporter gene activation. Since the experiments in Figures 3J and 4D involve different experimental conditions and show vastly different TOPflash responses to GSK3 inhibition I suggest to repeat the experiments shown in Figure 4E including LiCl and 12 h treatment time.

Response: As recommended by the reviewer, we repeated the TOPFlash luciferase assay in WT and PAWS1^{-/-} U2OS cells using LiCl for 12 h (**Response Figure 3**). Similar to our original data using GSK3 inhibition with CHIR99021 (Figure 4D), treatment of PAWS1^{-/-} cells with 20 mM LiCl for 12 h did not restore the Wnt-reporter activity to those seen in WT cells. Our use of the selective GSK3 inhibitor CHIR99021 instead of the non-selective LiCl in the original paper was due to criticisms on the use of LiCl in the first place.

This data, together with our experiments from Figure 9B, show that the most likely mechanism for inhibition of Wnt signalling in PAWS1^{-/-} cells appears to be due to the reduction in the nuclear accumulation of active β -catenin. A positive role for CK1 α in Wnt/ β -catenin signalling is not unprecedented. In Valle-Pérez et al., 2011 (Mol Cell Biol, 31: 2877-2888), the authors reported that depletion of CK1 α by siRNA led to an increase in β -catenin levels, yet there was no increase in transcriptional activity in Wnt3a treated cells compared to the untreated controls. This suggests that there is still some control of β -catenin downstream of GSK3 β activity. Therefore, these data as well as our data supports a model in which stabilized β -catenin is necessary but not sufficient for inducing Wnt-dependent transcriptional responses, and that a full transcription response requires an intact PAWS1::CK1 α complex. As we state in the manuscript, future phospho-proteomic analyses in WT and PAWS1^{-/-} cells with and without Wnt3A stimulation will potentially unravel the critical PAWS1:CK1 α mediators in Wnt signalling.

Overexpression of GSK3 in animal caps results in hyper-phosphorylated β -catenin. If PAWS1 acted upstream of GSK3, one would expect to see that co-expression of GSK3 with PAWS1 would still activate Wnt signalling, and thus induce *Siamois* expression. However, this is not the case. GSK3 overexpression blocked PAWS1-induced *Siamois* expression. Once β -catenin is phosphorylated and degraded, PAWS1 has no effect on it, corroborating with our hypothesis that PAWS1 is potentially involved in the nuclear translocation of the active β -catenin following Wnt stimulation. Importantly however, this hypothesis does not preclude from that possibility that CK1 α , perhaps through other FAM83 proteins, could still be involved in the regulation of β -catenin at Ser45 and the destruction complex.

Minor issues:

1. Page 10, line 26: Please refer to Fig 7E when describing the effects of PAWS1/CK1 α KD co-expression.

Response: We thank the reviewer for pointing this out. This has been corrected.

2. Fig 7E: I think the second column should be labelled with "CK1 α " (wildtype not KD).

Response: We thank the reviewer for pointing this out. This has been corrected.

Referee #2:

*This is a very interesting and well performed work.
The authors have in my view satisfactorily addressed all the comments of the reviewers.*

Response: We thank the reviewer for positive comments.

Referee #3:

The authors have conscientiously and thoroughly addressed the issues raised. This manuscript makes a significant contribution.

Response: We thank the reviewer for positive comments.

Response Figure 1: A & B. Two biological replicates of the experiment shown in the Main Fig. 9A. U2OS wild type and PAWS1^{-/-} cells were exposed to either Wnt3A or control medium for the indicated time points. Cell extracts were subjected to SDS-PAGE followed by Western blot analysis with the indicated antibodies using LiCor detection system. **C.** Plot of the quantification of the fold changes in phospho-β-catenin S45 over total β-catenin intensities in response to Wnt3A stimulation at the indicated time points, relative to the levels of the WT cells at t=0. The quantification was performed by using either the ImageJ software (densitometry – Fig. 9A) or with Licor (photon counts – A&B above) from three independent biological replicates (n=3; error bars represent ±SEM; 2-way ANOVA with multiple comparisons and Bonferroni's correction showed no significant differences between WT and PAWS1^{-/-} cells).

Response Figure 2: A&B. Two biological replicates of the experiment shown on Main Fig 9B, with the exception that only WT and PAWS1^{-/-} cells were used. U2OS wild type (WT) and PAWS1^{-/-} (KO) were exposed to either Wnt3A or control medium for 3h, followed by separation and preparation of cytoplasmic and nuclear fractions. The extracts were subjected to SDS-PAGE followed by Western blot analysis with the indicated antibodies using LiCor detection system. **C.** Plot of the quantification of the fold changes in active β-catenin intensities in the cytoplasmic and nuclear fraction relative to those seen in the cytoplasmic fraction of control WT U2OS cells (WT). The intensities of the active β-catenin bands in each fraction were quantified by using the Licor software. (n=3; error bars represent ±SEM; *p=0.0304; 2way ANOVA with multiple comparisons and Bonferroni's correction)

Response Figure 3: Relative TOPFlash luciferase activity of wild-type (PAWS1^{WT}) and PAWS1^{-/-} U2OS cells after treatment with either the conditioned medium (L-CM), WNT3A-conditioned medium (L3-CM) or 20 mM LiCl for 12 h. Data are normalized to Renilla-luciferase internal control (n=6; error bars represent ±SEM).

Thank you for the submission of your revised manuscript. Referee 1 is happy with the final revisions and we can therefore in principle accept your paper.

A few more minor changes are still needed:

- please provide up to 5 keywords
- figures 1A, 1E and EV1 need a scale bar
- in figures 1B,C and 3C,D the axis labelings are too small, please increase the font size. You can also find more information in our figure guidelines online:
http://www.embopress.org/sites/default/files/EMBOPress_Figure_Guidelines_061115.pdf
- please specify the statistical test used in the legend of figure 4B
- please define the p-value in figure 6B
- please add a callout for the movie in the manuscript text and explain what it shows
- the gel depicted in figure EV5A has a scratch, please clarify
- the gel depicted in figure 8 8th row seems to be cut, please clarify
- the black boxes shown in figures 2,3,5 and EV4 seem to be completely black. The actual images taken under the indicated conditions must be shown, and microscopy images usually have visible background. Please clarify.

To make it a little easier, I attach a word file to this email with comments on the figure legends. Please make the required changes in this document and send it back to us.

EMBO press papers are accompanied online by A) a short (1-2 sentences) summary of the findings and their significance, B) 2-3 bullet points highlighting key results and C) a synopsis image that is 550x200-400 pixels large (the height is variable). You can either show a model or key data in the synopsis image. Please note that text needs to be readable at the final size. Please send us this information along with the revised manuscript.

When you upload the new version of the manuscript you can bring forward all the old files and then only replace the files that need to be replaced.

REFEREE REPORTS

Referee #1:

Comments on manuscript EMBOR-2017-44807V3: "PAWS1 controls Wnt signalling through association with Casein Kinase 1 α " by Bozatz/Dingwell and colleagues.

I highly appreciate the authors' patience and their continuous efforts to address my concerns. The argumentation in their latest response is absolutely convincing and the additional experimentation removed all the remaining uncertainties that I had pointed out before (I wonder, though, why the authors did not incorporate these nice results into manuscript, but it is absolutely fine the way it is). Accordingly, I have no further comments. In fact, I expect that this study will receive a lot of well-deserved interest.

The authors addressed the minor editorial changes.

Corresponding Author Name: Gopal Sapkota and Jim Smith

Journal Submitted to: Embo reports

Manuscript Number: EMBOR-2017-44807-T